# A CD47-associated super-enhancer links pro-inflammatory signalling to CD47 upregulation in breast cancer

Paola A. Betancur[1], Brian J. Abraham[2], Ying Y. Yiu[1], Stephen B. Willingham[1], Farnaz Khameneh[1], Mark Zarnegar[1], Angera H. Kuo[1], Kelly McKenna[1], Yoko Kojima[3], Nicholas J. Leeper[3,4], Po Ho[1], Phung Gip[1], Tomek Swigut[1], Richard I. Sherwood[5], Michael F. Clarke[1], George Somlo[6], Richard A. Young[2,7] & Irving L. Weissman[1]

CD47 is a cell surface molecule that inhibits phagocytosis of cells that express it by binding to its receptor, SIRPα, on macrophages and other immune cells. CD47 is expressed at different levels by neoplastic and normal cells. Here, to reveal mechanisms by which different neoplastic cells generate this dominant 'don't eat me' signal, we analyse the *CD47* regulatory genomic landscape. We identify two distinct super-enhancers (SEs) associated with *CD47* in certain cancer cell types. We show that a set of active constituent enhancers, located within the two *CD47* SEs, regulate *CD47* expression in different cancer cell types and that disruption of *CD47* SEs reduces *CD47* gene expression. Finally we report that the TNF-NFKB1 signalling pathway directly regulates *CD47* by interacting with a constituent enhancer located within a *CD47*-associated SE specific to breast cancer. These results suggest that cancers can evolve SE to drive CD47 overexpression to escape immune surveillance.

[1] Institute for Stem Cell Biology and Regenerative Medicine, and Ludwig Center for Cancer Stem Cell Research and Medicine, Stanford University School of Medicine, Stanford, California 94305, USA. [2] Whitehead Institute for Biomedical Research, Cambridge, Massachusetts 02142, USA. [3] Department of Surgery, Division of Vascular Surgery, Stanford University School of Medicine, Stanford, California 94305, USA. [4] Department of Medicine, Division of Cardiovascular Medicine, Stanford University School of Medicine, Stanford, California 94305, USA. [5] Division of Genetics, Department of Medicine, Brigham and Women's Hospital and Harvard Medical School, Boston, Massachusetts 02115 USA. [6] Department of Medical Oncology & Therapeutics Research, City of Hope National Medical Center, Duarte, California 91010, USA. [7] Department of Biology, Massachusetts Institute of Technology, Cambridge, Massachusetts 02139, USA. Correspondence and requests for materials should be addressed to P.A.B. (email: betancur@stanford.edu) or to R.A.Y. (email: young@wi.mit.edu) or to I.L.W. (email: irv@stanford.edu).

CD47 is a cell surface glycoprotein that inhibits phagocytosis by binding to the extracellular region of SIRPα on macrophages. CD47 can be expressed on many normal cells in mice and humans[1]. Previous work in hematopoietic stem cells (HSCs) demonstrated that HSCs increase levels of CD47 expression when mobilized, particularly as they navigate through macrophage-rich environments. Once HSCs return to bone marrow, they downregulate CD47 (ref. 2). However, in cancer cells, CD47 transcript and protein expression is aberrantly upregulated, protecting the cancer cells from being recognized and cleared by macrophages[2,3]. Despite all this information, the mechanisms and upstream regulators responsible for increasing CD47 expression in cancer cells are still poorly understood.

Cis-regulatory regions or genomic enhancers are often referred to as 'switches' that regulate the transcription of a gene[4]. Recently the discovery of super-enhancers (SEs) has given more insight into the regulatory architecture of key genes that are highly expressed in a specific cell type, during a particular developmental stage or in disease. SEs are generally long stretches of DNA ($>20$ Kb) that contain clusters of enhancers, termed constituent enhancers[5]. SEs are also characterized by high levels of H3K27ac (an epigenetic modification that marks open chromatin)[6,7], which can exceed by one order of magnitude the levels found in a typical enhancer[8–10]. SEs were first described in embryonic stem cells as being associated with genes that control pluripotency[10]. More information has recently emerged that explains the role of SEs in upregulating master regulatory genes in various diseases, including atherosclerosis and cancer[8,11]. During atherosclerosis, stimulation by tumour necrosis factor-alpha (TNF-α) increases the binding of intracellular NFKB in cells associated with arterial blood vessels, creating SEs and a redistribution of BRD4 near genes that are upregulated during an inflammatory response[11]. In addition, in many cancers, SEs are acquired at critical genes that control and define the tumour cell identity[5,8,12].

Cancer cells of all types overexpress CD47 (ref. 3). Due to the critical role CD47 has in protecting cancer cells from phagocytosis, we hypothesized that SEs could be evolved by cancer or pre-cancer cells to induce overexpression of CD47, thus providing the cells with a selective advantage for growth and spread. Several studies have described CD47 transcriptional regulation in cancer cell lines by analysing the CD47 promoter region and the transcription factors interacting with it[13–16]. However, none of these studies discuss the role of distal enhancers or SEs in the regulation of CD47. To address this issue, we analysed the CD47 regulatory genomic landscape, to locate CD47 distal cis-regulatory regions (enhancers or SEs) and their upstream activators responsible for the upregulation of CD47 in cancer cells. Our goal is to identify alternative mechanisms and pathways directly upstream of CD47 that might be targeted to downregulate CD47 expression, thereby making cancer cells vulnerable to phagocytosis and immune clearance.

## Results

**SEs are associated with CD47 in breast and other cancers.** To better understand the regulatory genomic landscape of CD47, we analysed publicly available H3K27ac ChIP-Seq data for different cancer cell lines. By rank-ordering of enhancer regions based on H3K27ac enrichment, we discovered that T-cell acute lymphoblastic leukemia (T-ALL (RPMI18402, Jurkat and MOLT3)) diffuse large B-cell lymphoma (DLBCL (LY4)) and breast cancer (MCF7 and HCC1954) cell lines have SEs within ~200 kb of CD47 (Fig. 1a). Correlating this information with public microarray (Affymetrix U133 Plus 2.0) transcript expression data (Available at http://www.broadinstitute.org/ccle/home), we found that these cancer cell lines with SEs near CD47 are among those cancer types that express high levels of CD47 (Supplementary Fig. 1a). In contrast, cancer cell lines that have less SE signal around the CD47 locus (for example, some examples of lung, neuroblastoma and colorectal cancers; Supplementary Fig. 1b) express lower levels of CD47 (Supplementary Fig. 1a). In addition, our analyses of H3K27ac ChIP-Seq data indicated that CD47 is regulated by different sets of enhancers or SEs in different cancer cell types. For instance, the breast cancer cell lines HCC1954 and MCF7 have a downstream SE associated with CD47, while T-ALL and DLBCL cancer cell lines have either enhancers or SEs upstream of the gene (Fig. 1a).

Comparison of the CD47 enhancer landscapes of tumour cells versus corresponding normal (non-tumour) cells revealed that SEs were present only in the tumour cells. In T-ALL and breast cancer cells, CD47 is associated with an SE that is not present in CD3+ T cells or mammary epithelial cells, respectively (Supplementary Fig. 1c). This result is consistent with previous work suggesting that SEs are acquired by cancer cells[5]. However, in the case of breast cancer the downstream SE associated with CD47 is only present in two of seven tested breast cancer lines: MCF7 (Estrogen Receptor positive (ER+) and Progesterone Receptor positive (PR+) subtype) and HCC1954 (Human Epidermal Growth Factor positive (Her2+), ER−, PR− subtype) (Fig. 1a). Similarly, analyses of SEs in four patient derived-xenografted (PDX) breast tumour samples revealed that an ER+ PR+ breast cancer sample has the breast cancer SE associated with CD47 while the other three PDX breast tumour samples (triple negative: ER−, PR−, Her2−) do not (Fig. 1b).

**Identification of CD47 constituent enhancers.** SEs are comprised of multiple regions that function as transcriptional enhancers termed constituent enhancers. To find functional CD47 constituent enhancers within SEs that are sufficient to activate CD47 expression, we searched the CD47 genomic locus for highly conserved genomic regions across different species[17,18] that were also overlapped by H3K27ac and H3K4me1 (epigenetic hallmarks of open chromatin[7,19–21]) using ENCODE publicly available data and the UCSC genome browser (more details in the experimental procedure section). These analyses allowed us to predict 9 CD47-associated constituent enhancers (Supplementary Fig. 2a). To validate their function experimentally, we cloned each candidate CD47 enhancer (E1–9) into an EGFP reporter lentiviral construct containing the minimal (basal) promoter for the thymidine kinase (TK) gene[7]. To test each of the constructs, we transfected MCF7 and Jurkat cell lines because they have CD47 SEs (Supplementary Fig. 2b) and these lines express exceptional levels of CD47 protein (for example, almost 100 times and 10 times higher than the lowest expressing cell line, HepG2; Supplementary Fig. 2c). Since HepG2 cells express low levels of CD47 (Supplementary Fig. 2c,d) and lack CD47 SEs (Supplementary Fig. 1b), we used them as a negative control to confirm that reporter activity was not due to unspecific activation of the enhancers. We found that two of the CD47 enhancers (E5 and E3.2) had MCF7- and Jurkat-specific regulatory activity (Fig. 2a–c). First, E5, in the downstream CD47 SE seen in breast cancers (Fig. 1a,b), showed increased reporter activity specifically in the MCF7 breast cancer cell line (Fig. 2a). Further analysis of publicly available Paired-End Tag (ChIA-PET) data[22,23] confirmed that E5 and the downstream CD47 SE in MCF7 are connected by a DNA loop containing RNA Polymerase II, part of the complex of factors that are necessary to initiate transcription

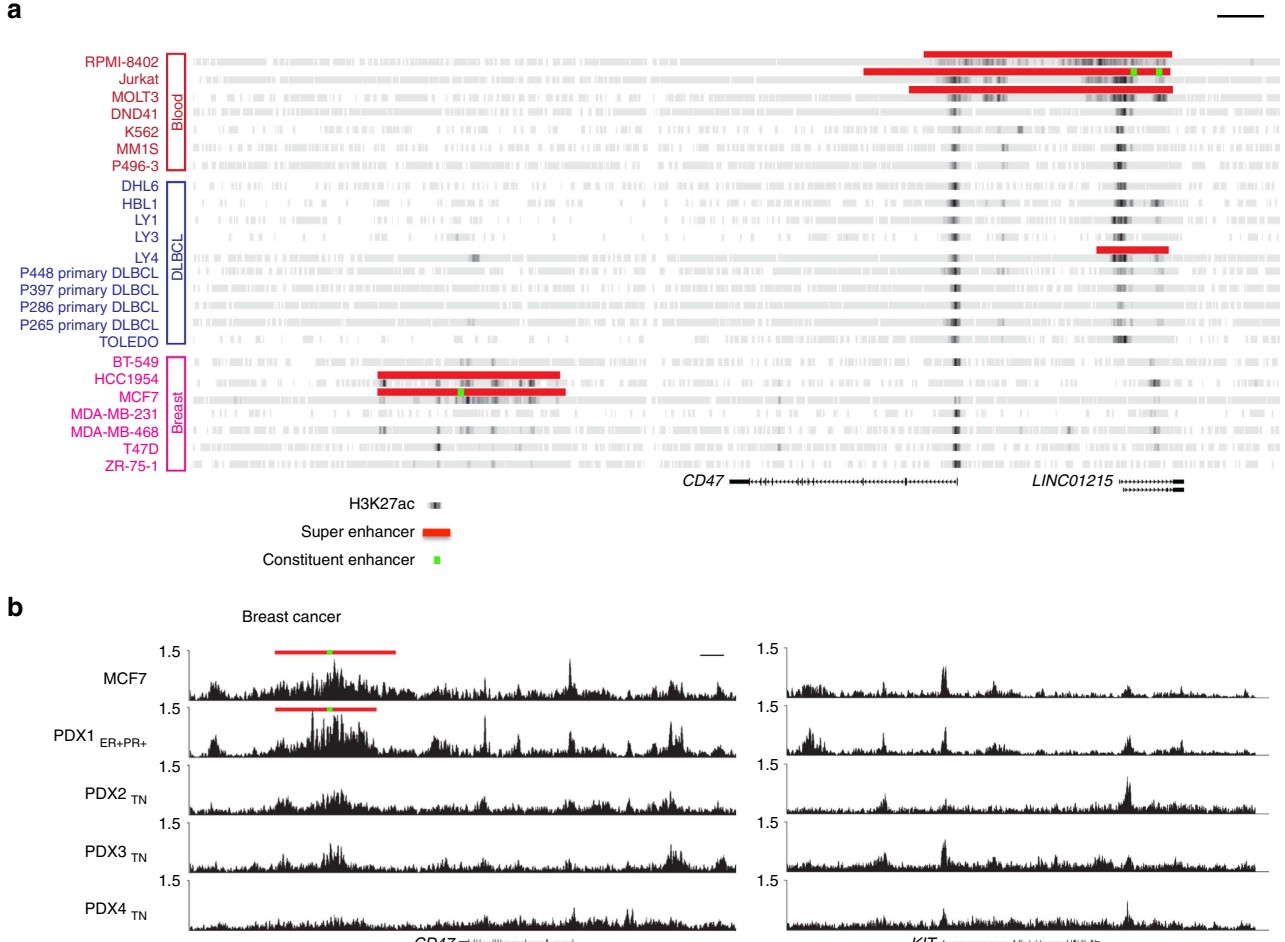

**Figure 1 | H3K27ac ChIP-Seq profiling reveals upstream and downstream *CD47* super-enhancers. (a)** Heat map representing H3K27ac enrichment (grey to dark grey) across different cancer lines shows that T-ALL lines (RPMI-8402, Jurkat and MOLT3), the DLBCL line, LY4 and breast cancer lines (HCC1954 and MCF7) have SEs (red lines on top) associated with *CD47*. Green blocks: represent functional E5, E3.2 and E7 constituent enhancers from left to right respectively. **(b)** H3K27ac enrichment (*y* axis) shows that a downstream SE (red line on top) is associated with *CD47* in an ER+ PR+ breast tumour sample (PDX1). Three other triple negative (TN = PR−, ER− and Her2−) breast tumour samples (PDX2-4) shown H3k27ac enrichment in the *CD47* locus, but these genomic regions do not qualify as SEs. Right panel: H3K27ac ChIP-Seq binding profiles show typical enhancers at the *KIT* gene for size comparison between SEs and typical enhancers. Green blocks: represents the functional E5 constituent enhancer. Scale bars:10 kb.

and are associated with promoter regions[22,24,25] (Supplementary Fig. 2e). Second, we found that E3.2, located within the upstream *CD47* SE (Fig. 1a), had increased reporter expression specifically in the Jurkat cell line (Fig. 2b). We also found that a third functional enhancer, E7, located within the upstream *CD47* SE (Fig. 1a), drove reporter expression in all of the cancer cell lines tested (Fig. 2d).

**NFKB1 binds to E5 and regulates CD47 in breast cancer cells.** To locate the upstream pathways that are responsible for the upregulation of *CD47* through the binding and activation of distal *cis*-regulatory regions, we focused first on finding candidate transcription factors that bind to the breast cancer specific constituent, E5. Thus, we performed a competitive protein-DNA binding assay, in which nuclear extracts from MCF7 and the negative control cell line, HepG2, were incubated with an array of 47 DNA probes (composed of consensus sites for well-characterized transcription factors) in the presence or absence of E5 DNA. We used this assay to search for transcription factors that were differentially bound to *CD47* E5 constituent enhancer in MCF7 and not in HepG2 cells. Our results showed that the transcription factors NFAT, NFKB, PPAR, SMAD, STAT3,

STAT5 and STAT6 from MCF7 and not from HepG2 nuclear extract bind to E5 (Fig. 3a). With the exception of NFAT and SMAD (which showed low binding to E5), we confirmed by quantitative real-time polymerase chain reaction (qPCR) that the candidate factors we identified in this binding assay are indeed expressed at different levels in the MCF7 breast cancer line (Supplementary Fig. 3a). Altogether, these data suggest that the transcription factors STATs 3, 5, 6, NFKB and/or PPAR could be key for the activation of CD47 expression by binding to E5 in MCF7 breast cancer cells. Next, we tested the regulatory capacity of each identified candidate factor on CD47 expression, by transducing MCF7 cells with short hairpin RNAs (shRNAs) to knockdown the expression of each candidate E5-binding transcription factor. We used qPCR to measure the efficiency of each knockdown (Supplementary Fig. 3b,c) and the effect on *CD47* gene expression. We observed that 72 h after shRNA transduction, expression of CD47 transcript and protein (measured by flow cytometry) was significantly reduced by NFKB1 (Fig. 3b,c) and PPARα (Fig. 3b, Supplementary Fig. 3d,e) shRNAs, when compared to control shRNA. On the other hand, shRNAs against STATs 3, 5 and 6 did not reduce *CD47* expression significantly (Fig. 3b). This information demonstrates that

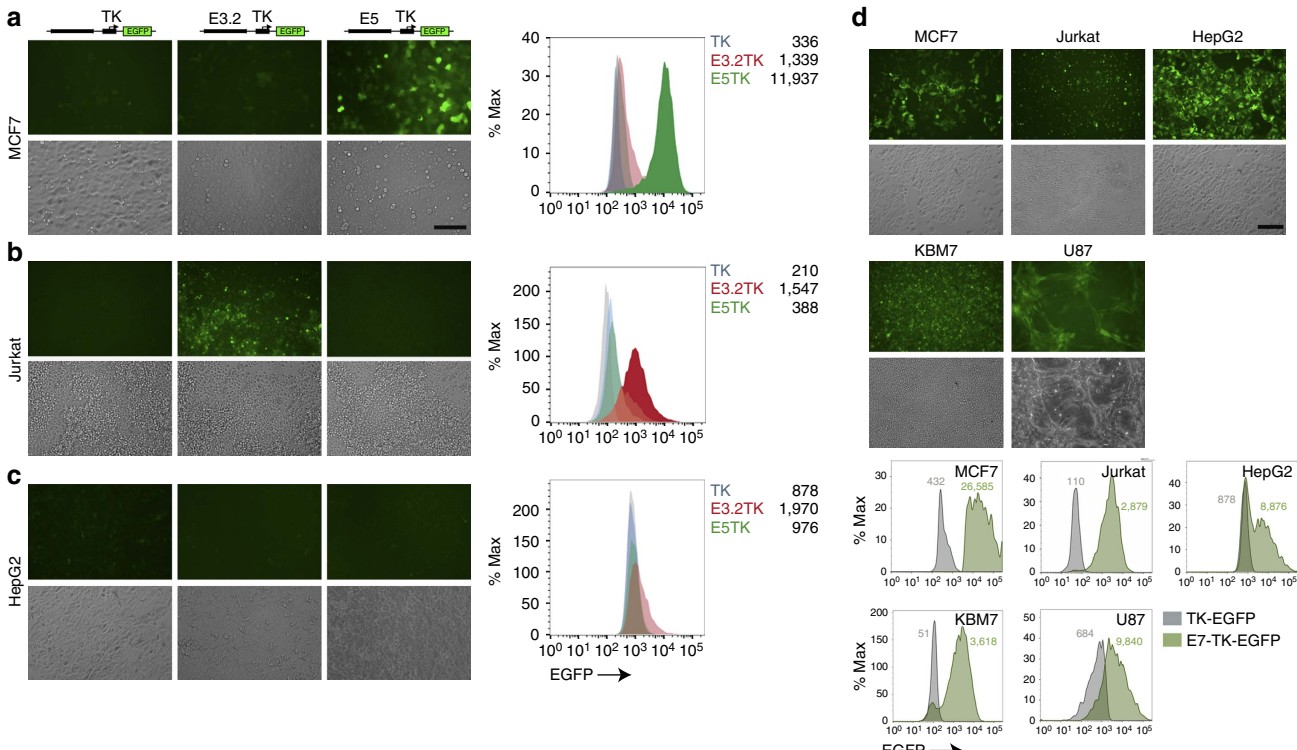

**Figure 2 | Identification and characterization of *CD47* SEs constituent enhancers.** (**a**) In MCF7 cells, specific EGFP reporter expression was activated by the E5 (right panel) *CD47* constituent enhancer region but not by E3.2 (middle panel) or others. (**b**) In Jurkat cells, specific EGFP reporter expression was activated by the E3.2 constituent enhancer (middle panel) but not by E5 (right panel) or others. (**c**) Neither E5 (right panel) nor E3.2 (middle panel) constituent enhancers activate reporter expression in HepG2 cells. Control cells transduced with the lentiviral cassette containing the thymidine kinase (TK) minimal promoter only (left panels, **a–c**). (**d**) E7 activates EGFP reporter expression in all the cancer cell lines tested, including the HepG2 negative control. Grey images (**a–d**) are corresponding bright field micrographs. Histograms show the mean of EGFP reporter signal measured by flow cytometry. Grey histograms (**a–d**) are the fluorescence minus one (FMO) controls. Scale bars: 100 μm.

NFKB and PPAR are involved in the regulation of CD47 in MCF7 cells and since knocking down NFKB1 had the strongest effect on *CD47* gene expression, we focus this study on the regulatory role of this transcription factor.

**A deletion containing an NFKB site inhibits E5 activity.** To confirm that the E5 constituent *CD47* enhancer carries functional binding sites that respond to NFKB1 binding, we computationally predicted the genomic locations of NFKB-binding sites within E5 to subsequently delete these regions. To approach this, we relied on analyses performed and published using a method termed Protein Interaction Quantification (PIQ)[26,27]. By using PIQ, we predicted the exact location of an NFKB1 binding site among binding sites for other transcription factors within E5 (Fig. 4a). To test if a region containing the predicted NFKB1 binding motif was indeed necessary for E5 regulatory activity, we performed two deletions within E5 and assayed the effect on EGFP reporter expression in MCF7 cells (Fig. 4a,b). Deleting 276 bp upstream of E5, a region that lacks the NFKB putative site (E5Δ276 bp), did not affect EGFP reporter expression when compared to the EGFP expression of the intact E5-TK-EGFP construct. On the other hand, a deletion close to the 3′ end (E5Δ400 bp), which included the NFKB1 putative binding motif (Fig. 4a), abolished EGFP reporter expression (Fig. 4b). A smaller version of the E5 (∼400 bp) containing the predicted NFKB1 binding motif (E5V400 bp) was not sufficient to activate EGFP reporter expression (Fig. 4a,b). Therefore, the region within E5 that contains the NFKB1 predicted motif is necessary but not sufficient for regulating the expression of CD47.

**NFKB1 knockdown reduces breast tumour size.** To test whether reducing CD47 expression by knocking down NFKB1 would have an effect on tumour growth and increase phagocytosis of the cancer cells, we infected MCF7-Luc cells with turboRFP-control shRNA or turboRFP shRNA against NFKB1 and then injected these and uninfected control cells into NSG (immunodeficient) mice. Prior to injections, we confirmed that the shRNAs did not affect cell viability (Supplementary Fig. 4a,b). The luciferase signal indicated that MCF7 cells infected with the NFKB1 shRNA initially formed tumours, but regressed over time (Fig. 5a); 6 weeks after injection, these tumours were smaller than those formed by the control MCF7 cells (uninfected or infected with turboRFP-control shRNA). In parallel, we measured CD47 levels of expression in the tumours after dissociation of the cells. CD47 expression levels were lower in the turboRFP-NFKB1 shRNA MCF7 tumours than in tumours formed by the control MCF7 cells (Fig. 5b). Lastly, we confirmed by *in vitro* assays that knocking down NFKB1 (via infection with a turboRFP shRNA-containing vector rather than an empty control vector) increases phagocytosis of MCF7 cells by macrophages. This effect is similar to the increase in phagocytosis observed when MCF7 cells infected with the control vector are treated with CD47-blocking antibody (clone Hu5F9-G4) before incubation with macrophages. Interestingly, the combination of NFKB1 knockdown by shRNA and CD47-blocking antibody dramatically increased phagocytosis by macrophages (Fig. 5c, Supplementary Fig. 4c).

**Disrupting the *CD47* breast cancer SE reduces *CD47* levels.** BRD4, a member of the bromodomain and extraterminal (BET)

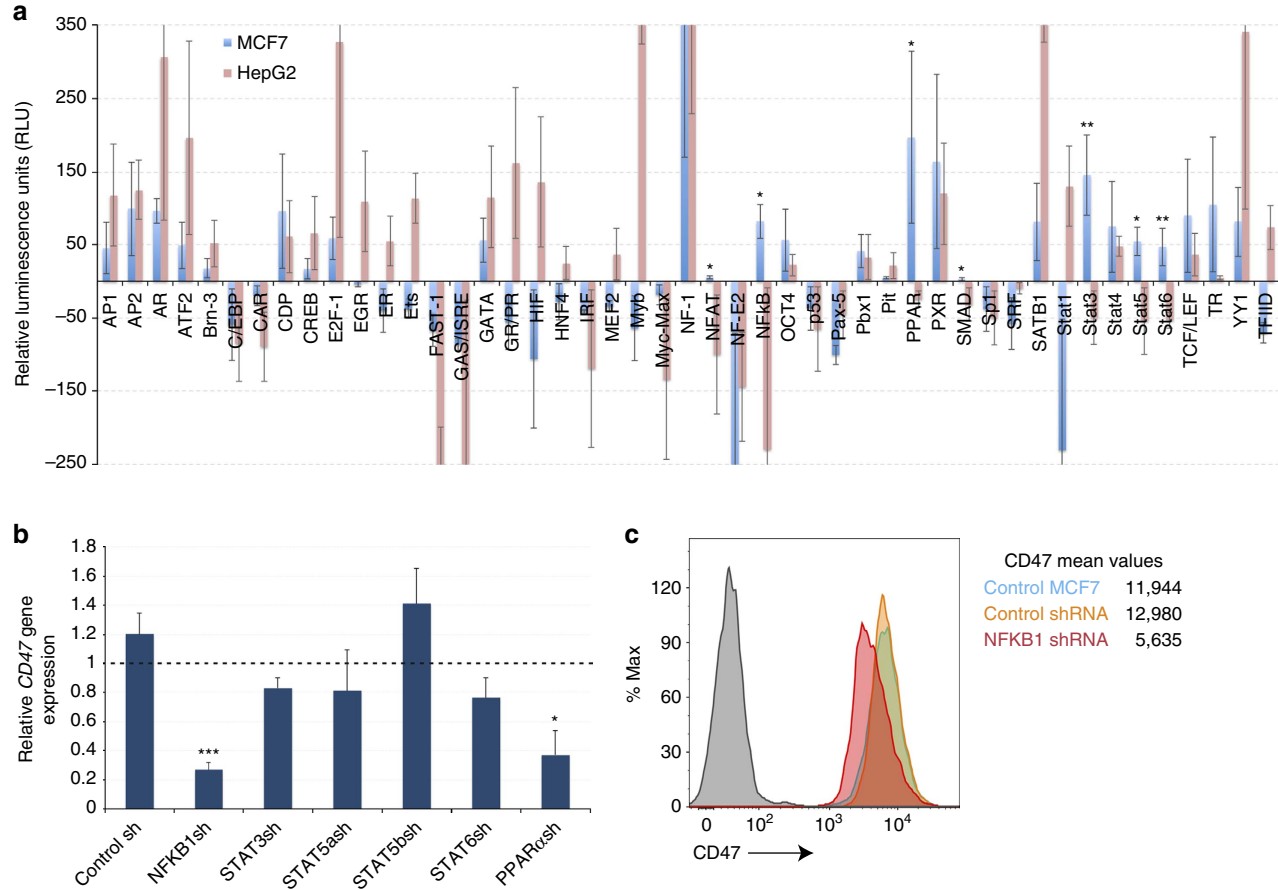

**Figure 3 | NFKB1 candidate transcription factor regulates CD47 expression in breast cancer cells.** (**a**) A protein-DNA binding profiling assay reveals that transcription factors NFAT, NFKB, PPAR, SMAD, STAT3, STAT5 and STAT6 bind significantly to E5 in MCF7 (blue bars) and not in HepG2 cells (red bars). Nuclear extract from MCF7 and HepG2 were incubated with a plate array containing oligos encoding consensus sites for well-characterized transcription factors. A competition assay was performed by incubating the nuclear extract with the E5 DNA fragment and the consensus sites oligos simultaneously. Final RLU (binding of each transcription factor to *CD47* E5) was calculated as follows: the average relative luminescence units (RLU) produced by the binding of each transcription factor to the consensus probes when outcompeted with the E5 DNA fragment (binding competition) was subtracted from the average RLU produced by the binding of each transcription factor to the consensus sites probes only (control). Thus, binding of transcription factors to the E5 DNA fragment and not to the consensus sites probes is represented by an increase in RLU while binding to the consensus sites probes and not to E5 is represented by a no change or decrease in RLU. The binding to E5 of each transcription factor obtained from the MCF7 nuclear extract was compared to the binding to E5 of each transcription factor obtained from HepG2 nuclear extract. $N = 4$ samples. Values represent mean ± s.d. Student's unpaired *t*-test for independent samples was performed. ** $P < 0.01$, * $P < 0.05$. (**b**) Knocking down NFKB1 and PPARα by shRNAs reduces CD47 gene expression more than knocking down other candidate transcription factors in the breast cancer cell line MCF7. $N = 5$ samples. Values represent mean + s.d. Student's unpaired *t*-test for independent samples was performed. *** $P < 0.005$, * $P < 0.05$. (**c**) Flow cytometry analyses show that CD47 cell surface protein levels are reduced after knocking down NFKB1 (red histogram) in MCF7 cells. Grey histogram is the FMO control.

subfamily of proteins, binds to hyperacetylated chromatin regions to facilitate rapid gene transcription by linking enhancers or promoters to the TEFb (transcription elongation factor) complex[28]. Previous studies on BRD4 have shown that: (i) BRD4 binds preferentially to SEs; (ii) disrupting BRD4 selectively affects the expression of genes that are associated with SEs; and (iii) the binding of NFKB to SEs recruits BRD4 to these sites[11,28–30]. Since our results show that NFKB1 binds to the E5 constituent enhancer within the breast cancer CD47 SE, we hypothesized that in breast cancer cells, *CD47* expression could be reduced when disrupting the *CD47* SE enhancing function by blocking the binding of BRD4 to SEs with the JQ1 inhibitor[29]. As expected, the treatment of MCF7 cells with JQ1 (1 μM) led to reduction in *CD47* transcript levels beginning on day 2 and reaching statistical significance on day 7 of treatment (Fig. 5d). This observation was confirmed when using other BRD4 inhibitors (I-BET151 and PFI-1), which reduced *CD47* expression dramatically at day 2 (Supplementary Fig. 5). However, these inhibitors seemed to be more toxic to the cells or be less specific than JQ1 as a decrease on number of cells was observed in the treated samples.

**Blockade of TNF-α decreases CD47 and increases phagocytosis.** Since our experiments demonstrated that NFKB1 activates the *CD47* E5 constituent enhancer and directly regulates *CD47* transcriptional expression, we investigated if the TNF inflammatory pathway, upstream of NFKB1, regulates CD47 expression in MCF7 cells and whether it contributes to the cancer cell's ability to avoid immunosurveilance. To approach this question, we first assayed the expression of a TNF-α receptor (TNFR1) on various cells. Flow cytometry analyses showed that TNFR1 expression levels were higher on the surface of the MCF7 cancer line than on MCF10 (a breast cancer cell line not considered tumorigenic) and the HepG2 (hepatoma) cancer line, which natively expresses low levels of CD47 (Fig. 6a). Next, we tested whether stimulation of cells with TNF-α ligand affected CD47

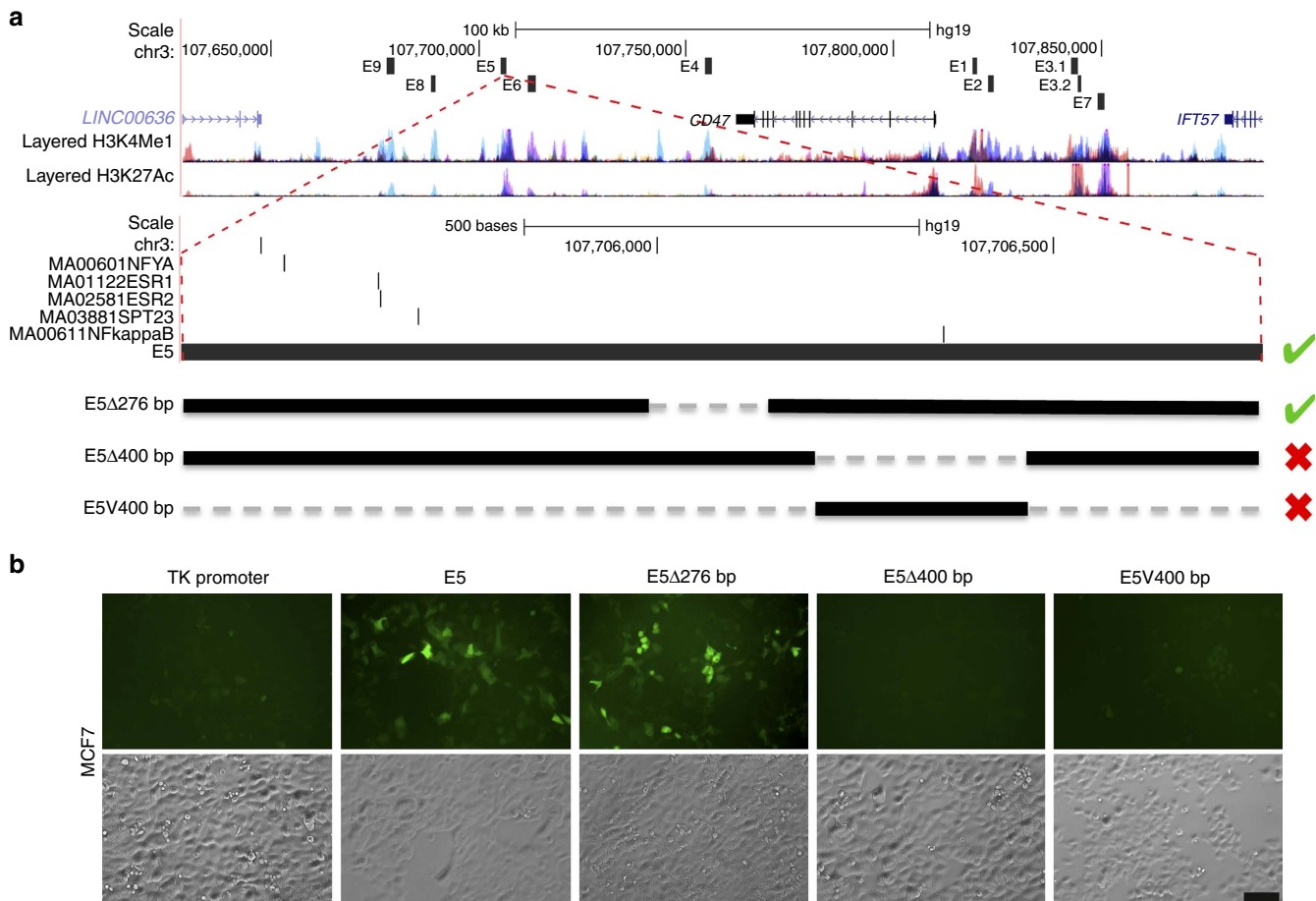

**Figure 4 | Regulatory activity of *CD47* E5 constituent is abolished by deleting a region containing an NFKB1 binding motif. (a)** Schematic representation showing the location for NFYA, ESRs, SPT23 and NFKB transcription factor binding sites (black bars) within *CD47* SE E5 constituent (black block), predicted by PIQ. H3K4Me1 and H3K27ac ChIP-Seq data publicly available for seven different cell lines (coloured peaks) mark regions of open chromatin. **(b)** Deletion of a fragment containing the NFKB1 binding motif (E5Δ400 bp) within *CD47* SE E5 constituent abolishes enhancer activity, while a 276 bp deletion excluding the NFKB1 motif (E5Δ276 bp) did not have any effect on E5 enhancer activity. However, a 400 bp fragment containing the NFKB1 binding motif (E5V400 bp) alone did not have enhancing activity. Scale bar: 100 μm.

gene and protein expression. Co-incubation with TNF-α for 48 h led to a significant increase in CD47 expression on MCF7 breast cancer cells (already CD47$^{Hi}$) but only a slight increase on nontumorigenic breast cancer MCF10 cells (CD47$^{Med}$) and the hepatoma cancer cell line HepG2 (CD47$^{Lo}$; Fig. 6b). To determine whether the increase in CD47 expression occurred at the transcript level, we performed qPCR at three different time points: 8, 24 and 48 h after addition of TNF-α. *CD47* transcript levels for MCF7 cells increased fourfold starting at 8 h and then decreased. For HepG2 an increase of transcript levels was observed later, at 24 h (Fig. 6b). Similarly a significant increase in *CD47* expression was observed in three out of four PDX breast tumour samples (PDX1 showing the greatest effect) and one primary patient breast tumour after treating the cells with TNF-α for 24 h (Fig. 6c, Supplementary Fig. 6a).

Upon TNF pathway stimulation, gene transcriptional activation can be achieved by activation of the Inhibitor of Nuclear Factor Kappa-B Kinase Subunit Beta (IKK2). IKK2 activation phosphorylates the inhibitor of NFKB (IkB) complex which normally binds and prevents NFKB from entering the nucleus of the cell. Phosphorylated IkB is degraded by the ubiquitination pathway and NFKB is released and allowed to enter the nucleus where it activates different target genes[31,32]. To confirm that TNF-mediated upregulation of CD47 on MCF7 cells is dependent on active signalling processes initiating NFKB translocation to the nucleus, we blocked IKK2 by using the TPCA-1 inhibitor and assayed the effect on CD47 expression. We found that CD47 expression does not increase after stimulating MCF7, MCF10 and HepG2 with TNF-α and treating with TPCA-1 simultaneously (Supplementary Fig. 6b). Moreover by western blot we show that in MCF7 cells phosphorylation and degradation of IkB occurs after TNF-α stimulation (Supplementary Fig. 6c). Next, to demonstrate that the CD47 upregulation observed after stimulating the TNF pathway is mediated by the activation of the *CD47* SE E5 constituent, we stimulated MCF7 cells carrying the E5-TK-EGFP with the TNF-α ligand and assayed EGFP gene and protein expression after 24 and 48 h. As a negative control we stimulated the cells with β-Estradiol (E2), due to: (i) the lack of functional ER binding sites within the *CD47* E5 constituent enhancer in MCF7 (Fig. 3a) and (ii) the inability of E2 to increase CD47 transcript and protein expression in treated MCF7 cells (Supplementary Fig. 6d). As expected, we observed a substantial increase in *EGFP* transcript 24 h after TNF-α stimulation and a significant increase in protein levels 24 and 48 h after treatment. No significant increase on EGFP or CD47 expression was observed after ER stimulation (Supplementary Fig. 6d). Lastly, we blocked BRD4 binding to the *CD47* breast cancer SE, by using the inhibitors I-BET151 and PFI-1 and we observed a reduction on TNF-α mediated *CD47* upregulation in MCF7 cells (Supplementary Fig. 6e).

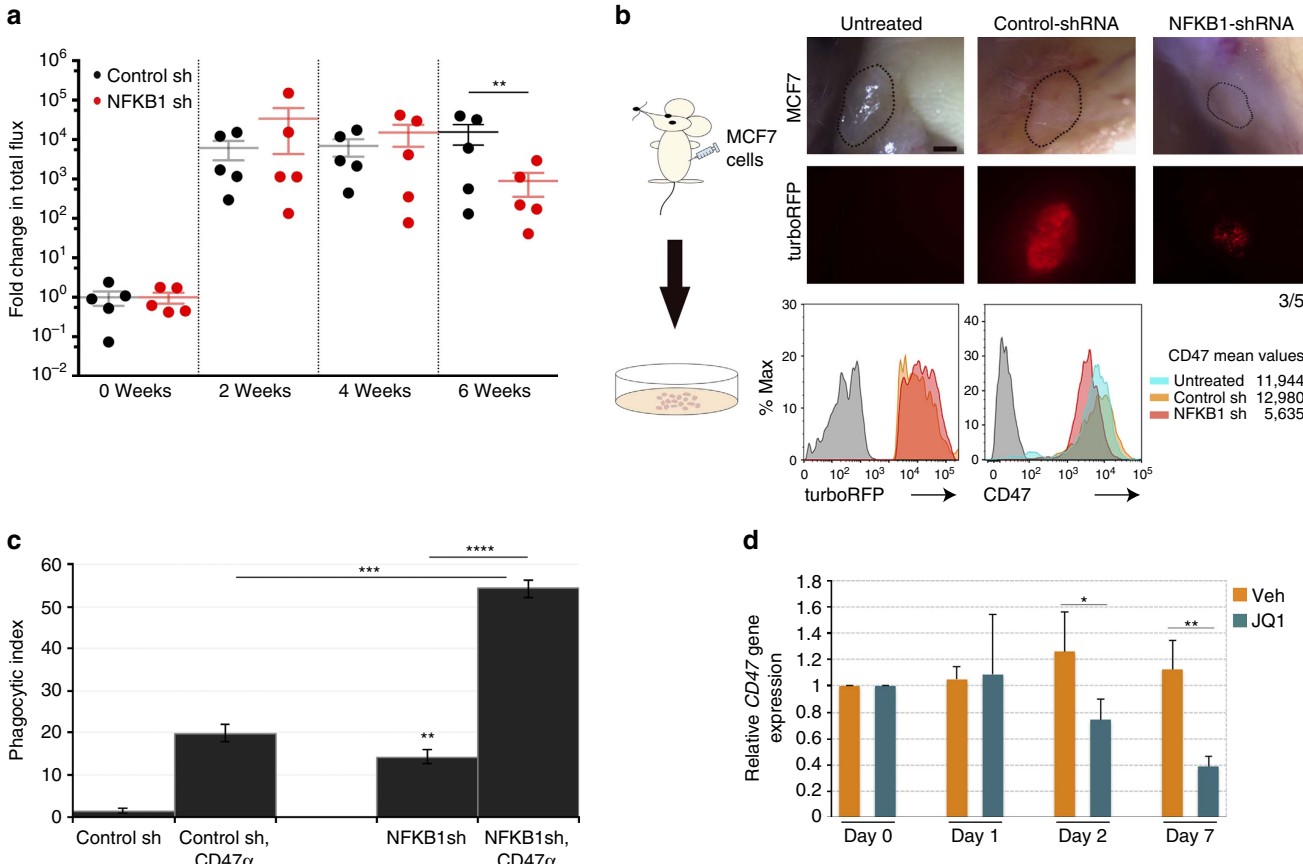

**Figure 5 | NFKB1 knockdown reduces CD47 expression, decreases tumour size and increases phagocytosis of breast cancer cells.** (**a**) NFKB1 shRNA reduces the size of breast tumours derived from MCF7-Luc in xenotransplants (red dots). $N = 5$ samples. Values represent mean ± s.e.m. Student's unpaired $t$-test for independent samples was performed. ** $P < 0.01$. Black dots: Control shRNA treated breast tumours derived from MCF7-Luc in xenotransplants. (**b**) Upper panels: Images of tumours 6 weeks after injection of MCF7 cells that were: untreated (control; left); infected with turboRFP-control shRNA (middle); or infected with turboRFP-NFKB1 shRNA (right). NFKB1 knockdown led to smaller tumour sizes. Scale bar: 500 μm. Lower panels: flow cytometry analysis of cells from dissociated tumours that had grown for 6 weeks. Left panel: TurboRFP (reporter) expression levels are similar in cells treated with turboRFP-control shRNA (yellow) versus turboRFP-NFKB1 shRNA (pink). Orange histogram shows overlap. Right panel: CD47 protein levels are lower on cells in tumours generated from MCF7 cells with NFKB1 knockdown (red) versus those without knockdown (controls with no shRNA (turquoise) or with control shRNA (orange)). CD47 expression is slightly higher in MCF7 cells infected with control shRNA than in uninfected MCF7, because any intracellular infection leads to an increased CD47 expression. Grey histograms are the FMO controls. (**c**) Phagocytic index of MCF7 cells is increased over control levels (empty vector lacking shRNA) by anti-CD47 blocking antibody (CD47α), NFKB1 shRNA infection, and more so by NFKB1 shRNA infection followed by anti-CD47 treatment. $N = 3$ samples. Values represent mean ± s.d. Student's unpaired $t$-test for independent samples was performed. ** $P < 0.0025$ when compared to Control shRNA, *** $P < 0.0010$, **** $P < 0.0005$. (**d**) Inhibiting the binding of BRD4 to SEs by using JQ1 (1 μM) reduces *CD47* expression over time in MCF7 breast cancer cells. $N = 3$ samples. Values represent mean + s.d. Student's unpaired $t$-test for independent samples was performed. * $P < 0.05$, ** $P < 0.01$.

Since stimulating MCF7 cells with TNF-α increases CD47 expression, we then blocked the TNF pathway as a possible translational approach to reduce CD47 expression and aid phagocytosis of the cancer cells. To do so, we used infliximab (trademark, Remicade), a chimeric monoclonal antibody that binds to TNF-α and prevents its binding to cellular TNF receptors[33]. This blocking antibody is already used for the treatment of inflammatory diseases such as ulcerative colitis, psoriasis, arthritis and Crohn's disease[33]. Flow cytometry analysis of CD47 expression confirmed that blocking TNF-α with infliximab reduces the expression of CD47 at the cell surface, with or without stimulation with TNF-α (Fig. 6d). The reduction of CD47 observed without TNF-α stimulation could be in part due to the fact that MCF7 breast cancer cells also express transmembrane TNF-α at very low levels[34], and thus, TNF-α could be activating the TNF pathway through an autocrine and/or paracrine interaction with the receptor. Moreover,

a greater reduction of CD47 expression was observed when we combined the infliximab antibody treatment with the shRNA against NFKB1 (Fig. 6d), implicating NFKB as a transducer of CD47 upregulation. *In vitro* phagocytosis assays demonstrated that blocking the TNF pathway with infliximab increases phagocytosis of the cancer cells, and the effect is greater if infliximab is combined with an anti-CD47 blocking antibody (clone Hu5F9-G4). A greater increase in phagocytosis was observed when the NFKB1 shRNA-MCF7 line was treated with both blocking antibodies (Fig. 6e). Surprisingly, we also observed an unexpected increase in phagocytosis of MCF7 cancer cells after stimulation with TNF-α ligand. Since (i) MCF7 cells are known to be sensitive to soluble TNF-α-induced cell death[34] and (ii) TNF-α has been previously reported to induce long-term tumour regression[35,36], we speculated that TNF-α could be upregulating CD47 and apoptotic or prophagocytic genes in MCF7 cells simultaneously. Indeed, we observed a slight increase in the expression of the prophagocytic

signal, calreticulin, on the surface of MCF7 cells after stimulation with TNF-α (Supplementary Fig. 7)[37].

## Discussion

We have previously found that antibodies that block the interaction of CD47 with macrophage surface SIRPα release the macrophage to phagocytose and destroy the cancer cells. However, important questions that have yet to be answered include: (1) What is or are the mechanism(s) by which CD47 becomes overexpressed in cancer cells; and (2) What are the upstream regulators involved in the overexpression of CD47?

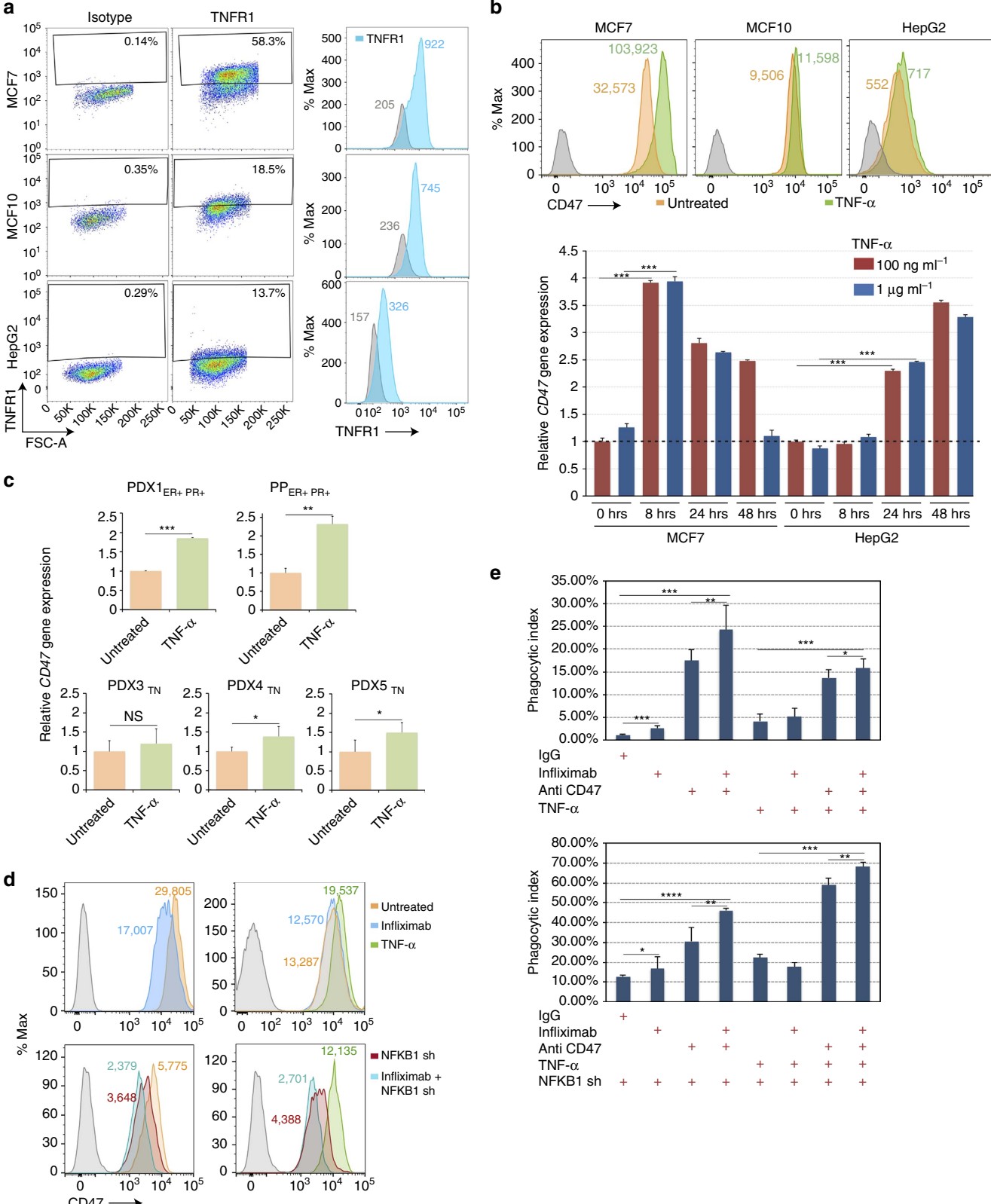

To answer the first question, we performed a genomic regulatory analysis to locate the *CD47* cis-regulatory regions or enhancers active in cancer cells. Our *in silico* analyses suggest that cancer types characterized by the presence of *CD47* SEs have high levels of *CD47* transcript. Thus, we believe that activation of SEs is a mechanism used by certain cancers, such as breast cancer, T-ALL and DLBCL, to obtain high CD47 levels. However, to add to the complexity of *CD47* regulation by SEs, we found that not all the breast cancer subtypes have associated SEs with *CD47*. Breast cancer is divided into five subtypes with distinguishable features and distinct clinical outcomes, based on the expression of ER, PR, HER2, CK5/6 and EGFR markers[38]. So far we found that two breast cancer cell lines (HER2 or ER+ PR+) and one PDX breast tumour (ER+ PR+) have SEs associated with *CD47*. Perhaps SEs are subtype specific or patient specific. Now that the tools to perform SEs analyses in patient samples are becoming more available, it would be important to address this possibility by analysing many more breast cancer subtype samples for the presence of specific SEs.

While screening for functional *CD47* cis-regulatory regions (enhancers or constituent enhancers within SEs), we discovered that (i) a functional enhancer (E7), located upstream *CD47*, had enhancing activity that was common to all the cancer cell lines tested; (ii) two functional constituent enhancers, E5 present within a downstream *CD47*-associated SE and E3.2 present within an upstream *CD47*-associated SE, are to date only active in MCF7 breast cancer cells or Jurkat T-ALL cells respectively. This information suggests that even though *CD47* gene expression can be regulated in a broader range of cancer cells by one common enhancer, certain constituent enhancers located within *CD47* SEs seem to have a more tumour type-specific regulatory activity. Enhancers with such specific roles have been previously described to occur during development[39–42]. On the other hand, other candidate *CD47* enhancers tested did not have regulatory activity. We place the *CD47* candidate E6 enhancer in this category, even though it is located within the *CD47* downstream breast cancer SE. It is possible that these enhancers that fail to increase reporter expression are not functional, have specific regulatory activity to other cells types not tested, or they function as cooperative enhancers[5] and their regulatory activity alone is not sufficient to increase transcriptional expression, as previously described.

In this study, analyses of H3K27ac enrichment around the *CD47* locus indicated that SEs are present in T-ALL and breast cancer cells but not in their normal counterparts, CD3+ T cells and human mammary epithelium (Supplementary Fig. 1c). This suggests that cancer cells with *CD47* SEs have newly acquired these SEs, which perhaps increase the probability of high CD47 expression by simultaneously recruiting multiple constituent enhancers, including the constituent enhancers that have specific

regulatory activity. Supporting this idea, *in silico* analyses of H3K27ac enrichment around the *CD47* locus in different normal cells from various tissue types showed that in normal cells analysed, including, embryonic cells and cells obtained from tissue such as liver, ovary and lung, CD47 seems to be regulated by typical enhancers broadly distributed upstream and downstream of the *CD47* genomic locus. Although it appears that cells obtained from the chorion and amniotic extraembryonic tissues have *CD47* SEs (due to H3K27ac enrichment), their *CD47* genomic regions do not qualify as such based on the ranking of enhancers in order of H3K27ac binding density (Supplementary Fig. 8)[10,30,43]. Our studies also show that SEs are indispensable for the upregulation of *CD47*. In MCF7 cancer cells, we observed that disrupting SEs by using several BRD4 inhibitors reduces *CD47* expression. The ∼60% reduction on *CD47* transcript observed at day seven after inhibiting the binding of BRD4 to the *CD47* SE indicates that perhaps transcription factors such as HIF-1, that bind to the *CD47* promoter region, could upregulate CD47 expression independently of the downstream *CD47* E5 constituent enhancer in breast cancer[16]. In Jurkat cancer cells we also observed a great *CD47* reduction (60%) 48 h after JQ1 treatment (Supplementary Fig. 9a). *CD47* downregulation in this case may be caused by disrupting the SE located upstream of *CD47* and at the same time, the SE associated with the *Myc* gene, a transcription factor that also can regulate *CD47* in Jurkat[13].

In addressing the second issue of identifying upstream regulators of CD47 expression, we used the PIQ computational predictions together with a binding assay array to identify binding motifs for transcription factors directly modulating CD47 expression. We found that in MCF7 cells perturbing the transcription factor PPARα reduces CD47 expression by 60% suggesting the involvement of a metabolic pathway in the regulation of CD47 at least in breast cancer[44,45]. Such role is currently being explored. More importantly, our study shows that the transcription factor NFKB1 binds to and is necessary for the activation of the *CD47* constituent enhancer E5. Thus, NFKB1 binding to *CD47* E5 works as a switch for the regulation of CD47 expression. To turn off this regulatory switch, we performed an shRNA knockdown to inhibit the NFKB1 regulatory input on CD47. The turning off of this 'switch' resulted in a 70% reduction of CD47 levels, which promoted phagocytosis of MCF7 cancer cells *in vitro* and reduced tumour size in xenotransplantation models. Our data support reports where NFKB family members have been found to improve cancer cell survival and to likely play crucial roles in cancer initiation, progression and moreover, resistance to chemotherapy and hormonal therapy in breast cancer[32,46–48]. Finding that NFKB1 is an upstream CD47 regulator also corroborates the utility of PIQ predictions in determining functional upstream factors regulating the

**Figure 6 | The TNF inflammatory pathway affects phagocytosis of breast cancer cells through the regulation of CD47 expression.** (**a**) Flow cytometry analysis showing TNFR1 expression in the breast cancer line MCF7 (CD47[Hi]), non-tumorigenic breast line MCF10 (CD47[Med]) and HepG2 (CD47[Lo]) hepatoma cancer line shows that TNFR1 expression is higher in the MCF7 breast cancer cells. Mean values of TNFR1-FITC fluorescence are shown in blue to the right of each grey histogram. Grey histograms represent FMO controls. (**b**) Upper figure: A greater increase in CD47 expression is observed in MCF7 when compared to MCF10 or HepG2 cell lines after stimulation with TNF-α (green histogram). Mean values of CD47-APC are shown next to each histogram. Lower figure: Increase in *CD47* transcript expression begins to occur after 8 h in MCF7 cells and after 24 h in HepG2 cells upon TNF-α stimulation. N = 3 samples. Values represent mean + s.d. Student's unpaired *t*-test for independent samples was performed. ***P < 0.001. (**c**) Treating patient-derived xenografted (PDX) and primary patient (PP) breast tumour cells with TNF-α significantly increases *CD47* gene expression in four out of five breast tumour samples. Values represent mean + s.d. Student's unpaired *t*-test for independent samples was performed. ***P < 0.001, **P < 0.01, *P < 0.1. NS = not significant. ER+ PR+ = oestrogen and progesterone positive; TN = triple negative. (**d**) Upper panels: Blocking TNF-α with infliximab reduces CD47 expression regardless of TNF-α stimulation. Lower panels: Blocking TNF-α with infliximab while knocking down NFKB1 causes a greater reduction on CD47 expression regardless of TNF-α stimulation. (**e**) Treating MCF7 with infliximab increases phagocytosis of the cancer cells, and, to a greater extent, if infliximab is combined with CD47 blocking antibody. A more pronounced increase in phagocytosis of the cancer cells was observed for MCF7 cells that have NFKB1 knockdown and were treated with the infliximab and CD47 antibodies combined. N = 6 samples. Values represent mean + s.d. ****P < 0.0005, ***P < 0.001, **P < 0.05, *P = 0.15 (upper figure), *P = 0.20 (bottom figure).

expression of a particular target gene in a cell. Other PIQ analysis predicts the location of binding motifs for NFKB family members within the functional *CD47* E7, which is active in other cancer cell lines besides MCF7 (Supplementary Fig. 9b). This suggests that inflammation and the NFKB1 'switch' could also be involved in the upregulation of CD47 expression in other cancer types, such as T-ALL, through the activation of a different *CD47 cis*-regulatory region.

It is well known that the NFKB1 transcription factor is downstream from the TNF inflammatory pathway[32]. Thus, we also investigated the role of the TNF pathway. Our experiments show that stimulating cell lines that express TNFR1 receptor with TNF-α forces the cells to upregulate CD47 protein levels. Moreover, the CD47 increase was greatest in the MCF7 breast cancer cell line compared to other lines, even though MCF7 cells already express high CD47 levels. *CD47* transcripts were also more rapidly upregulated in MCF7 than in the low-CD47-expressing, HepG2 cancer line. Perhaps the prominent CD47 protein increase and rapid transcript upregulation occurring in MCF7 cancer cells is due to the presence of the CD47 SE characterized by hyper H3K27ac (which is absent in HepG2). Thus, we speculate that open chromatin around the *CD47* locus in the MCF7 breast cancer cell line is more accessible and sensitive to the binding of NFKB1 after TNF-α stimulation, making *CD47 cis*-regulatory regions more responsive to inflammation in these cancer cells than HepG2 cancer cells for instance. Overall our data demonstrate for the first time that the TNF inflammatory pathway upregulates CD47 expression in breast cancer cells. Our results also validate recent results observed in a hepatoma cell lines resistant to Sorafenib, where it was proposed that resistance to the drug treatment was associated with high CD47 protein levels achieved by the binding of NFKB2 subunit to the CD47 promoter upon TNF-α stimulation[15]. In a different study, it was also shown by our group that regulation of CD47 by the TNF pathway occurs in vascular smooth muscle cells (SMCs) during atherosclerosis, where TNF-NFKB1 was found to interact with the CD47 promoter[49].

Lastly in this report, we also demonstrate that blocking the TNF pathway, by using the antibody infliximab, increases phagocytosis of MCF7 breast cancer cells mediated by a reduction on CD47 expression. We show that a further increase in phagocytosis was achieved when treating the MCF7 cells with the CD47 blocking antibody combined with infliximab or NFKB1 shRNA or both. Perhaps this increase in phagocytosis is a result of indirect regulation by NFKB1 or TNF-α on genes other than CD47, encoding antiphagocytic or prophagocytic signalling proteins. Thus, treating MCF7 cells with the CD47 blocking antibody while perturbing the TNF pathway creates the right balance of prophagocytic and antiphagocytic signals necessary to recruit macrophages to engulf the cells. Supporting this, we observed an increase in calreticulin after TNF-α treatment[37,50]. This information altogether suggests that combining treatments, for instance the infliximab antibody with the CD47 antibody, could be considered as a therapy to increase engulfment of cancer cells by macrophages. Such combinatorial treatments have been shown to increase the efficacy of antibody treatment by blocking in parallel different mechanisms that cancer cells use to avoid being cleared by the immune system. For instance, recently it was demonstrated that the blocking of programmed death-ligand in tumour cells to recruit the adaptive immune system is important to potentiate the recruitment of the innate immune system (through the disruption of CD47–SIRPα interaction) to target cancer cells[51]. Our study describes in detail a mechanism that through the downregulation of CD47 increases the recruitment of the innate immune system. Thus, investigating if the blocking of

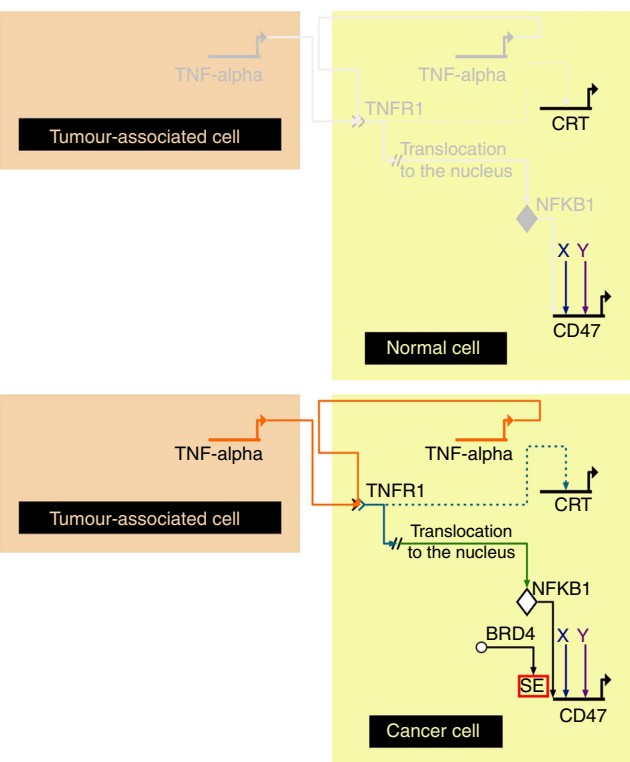

**Figure 7 | Schematic model representing how *CD47* transcriptional expression is regulated during inflammation in cancer.** The TNF pathway activates the translocation to the nucleus of NFKB1. Upstream unknown factors, bind to closed DNA, making SE constituents accessible to NFKB1 binding, which in turn activate SEs and recruit BRD4 to these SEs promoting the interaction between the distal SE(s) and the promoter of the gene to initiate transcription. Dotted lines indicate indirect interactions while solid lines indicate direct interactions.

the TNF pathway also has an effect in the recruitment of the adaptive immune system (perhaps through the regulation of programmed death-ligand) is important.

In summary, CD47 is regulated by sets of different enhancers and SEs in different cancer types. We found that particularly in breast cancer cells, CD47 expression is increased by an inflammatory pathway whereby the NFKB1 transcription factor directly regulates the expression of the gene by binding to a specific constituent enhancer within an active breast cancer SE (proposed mechanism, Fig. 7). Furthermore we showed, by perturbing the TNF-NFKB1 pathway, that reduction of CD47 expression leads to increased phagocytosis of MCF7 breast cancer cells. It is critical to understand the events regulating *CD47* gene expression at the transcriptional level as this has the potential to lead to alternative cancer gene therapies targeting the upstream group of transcription factors, epigenetic modifiers and molecular pathways responsible for CD47 upregulation, thus allowing for the recruitment of the host innate immune system to specifically recognize, target and destroy the cancer tumorigenic cells.

## Methods

***In silico* identification of super-enhancers.** Publicly available ChIP-Seq data targeting H3K27ac were downloaded from the GEO database (http://www.ncbi.nlm.nih.gov/geo/) and processed to identify SEs in a panel of human cell types (accession numbers for data sets used are listed in Supplementary Table 1). Raw reads were aligned to the hg19 revision of the human reference genome using bowtie[52] with parameters -k 2 -m 2 -sam -best and -l set to the read length. Constituent enhancers were identified using MACS (ref. 53) twice with parameters -p 1e9 and -keep-dup 1 or -p 1e-9 and -keep-dup all. The union of both peak calls was used as input for ROSE (https://bitbucket.org/young_computation/rose/).

ROSE separated SEs from typical enhancers with parameters -s 12,500 -t 2,000. Wiggle files for display of H3K27ac signal were created using MACS with parameters -w -S -space = 50 -nomodel -shiftsize = 200. Displays were created using the UCSC Genome Browser[54]. H3K27ac ChIP-Seq data obtained from four PDX samples were also processed using the same parameters described above.

**In silico analysis and CD47 enhancer constructs generation.** For many decades the study of enhancers has been based on the principle that non-coding genomic sequences that are conserved through evolution must be carrying important gene expression regulatory functions[55,56]. To predict CD47 functional cis-regulatory regions (enhancers) within SEs, we used The UCSC genome browser (https://genome.ucsc.edu) to identify highly conserved non-coding genomic regions across different species. To approach this, we aligned the human annotated CD47 genomic locus (~100 kb) to the CD47 annotated genomic locus available for a group of other vertebrate species including mouse, dog, armadillo, opossum, platypus, chicken, Xenopus, zebrafish and lamprey. In our analyses, the CD47 non-coding genomic sequence starts to diverge substantially in opossum and platypus, thus allowing for the visualization of isolated non-coding genomic patches of DNA that remain conserved across the different species compared (Supplementary Fig. 2a, black bars). To identify highly evolutionary conserved genomic regions also marked by H3K27ac and H3K4me1 epigenetic modifiers (well characterized for marking open chromatin), we looked at ENCODE H3K27ac and H3K4me1 ChIP-Seq profiles within the CD47 genomic locus available for a panel of seven human cell types (Supplementary Fig. 2a, coloured peaks) and aligned them to the non-coding genomic CD47 sequences that were already aligned to find evolutionary conservation across different species (Supplementary Fig. 2a, black bars). Putative CD47 enhancers (that is, non-coding regions highly conserved and H3K27ac and H3K4me1 enriched) identified in silico (E1-E9) were amplified by PCR from human bacterial artificial chromosome (BAC) libraries (CHORI: Clone # RP11-25F15 or RP11-1008K13) and cloned into a lentiviral reporter construct (Sin-minTK-eGFP) containing a minimal TK promoter (generously provided by Alvaro Rada-Iglesias). Fragments were cloned using restriction enzymes XhoI or BamHI at the 5′ end and MluI or XhoI at the 3′ end. Primer sequences are listed in Supplementary Table 2.

**Testing of regulatory regions.** Lentiviruses carrying the constructs containing CD47 putative enhancers were used to transduce MCF7, Jurkat, HepG2, U-87 and KBM7 cancer cell lines. MCF7, HepG2, U-87 cell lines (ATCC) were seeded overnight in a six-well plate at $5 \times 10^5$ cells per well. Then, cells were transduced for 2 days. Enhanced or decreased regulatory activity was measured 4 days after transduction by analysing EGFP fluorescence activity using fluorescence microscopy or flow cytometry. Jurkat (ATCC) and KBM7 (kindly provided by Dr Calo-Velazquez) cancer cell lines were seeded in a 24-well plate at $1 \times 10^6$ cells per well. Then, cells were transduced by spin infection (cycle time: 45 min, speed: 2,000 r.p.m. and temperature: 25 °C). Enhanced or decreased regulatory activity was measured 4 days after transduction. No additional cell authentication or mycoplasma contamination was performed in the cell lines used in this study.

**Locating the functional region within CD47 E5.** To locate the minimal region containing functional binding motif(s) within CD47 E5 (1,300 bp), we first predicted regions within E5 that had motifs with binding capacity by using published PIQ analyses. PIQ is an analytical method that combines existing ENCODE K562 DNase I hypersensitivity data[27] with aggregate transcription factor specific scores obtained from JASPAR, UniPROBE and TRANSFAC to predict the probability of occupancy for a given candidate binding site in the genome by using standard parameters previously described[26]. The published resulting calls for putative binding motifs in the genome were visualized using the custom track feature from the UCSC genome browser (https://genome.ucsc.edu), with a score threshold of 800.

Based on these analyses, we deleted two regions from E5 (E5Δ276 bp and E5Δ400 bp) by fusion PCR and generated a smaller CD47 E5 construct (E5V400 bp). The set of primers used for the generation of these new constructs were as follows:

E5Δ276 bp (~ 276 bp deletion)
_F′: ATAAGGATCCTCAGACTTAGTTTGTAGATGG
_R′: ATAACCTTCAAACAGGCCTGCTTGATTGGGAGAAAACACC
_F′: CAGGCCTGTTTGAAGGTTAT
_R′: TAATACGCGTATAACACAGGGAATAGAAGC
E5Δ400 bp (~ 400 bp deletion)
_F′: ATAAGGATCCTCAGACTTAGTTTGTAGATGG
_R′: CTTGATTGGGAGAAAACACCTGCATCTTGTATGTGGTTGG
_F′: GGTGTTTTCTCCCAATCAAG
_R′: TAATACGCGTATAACACAGGGAATAGAAGC
E5V400 bp (~ 400 bp fragment)
_F′: AATTGGATCCTCCCAACCACATACAAGATG
_R′: ATATACGCGTCTTGATTGGGAGAAAACACC

MCF7 cells were infected with each new CD47 E5 construct and the effect on EGFP reporter expression assayed by fluorescence microscopy 4 days after infection.

**Phagocytosis in vitro assay.** In vitro macrophages were induced and cultured for 7 days as previously described[2]. Macrophages were collected from a GFP murine line. Target cells (MCF7 cells) transduced with turboRFP-NFKB1 shRNAs or turboRFP alone (control) were subjected to $10 \, \mu g \, ml^{-1}$ of CD47 blocking antibody (Hu5F9-G4) for 30 min at 37 °C. Target cells with the different conditions were then plated with macrophages in a low-attachment 96-well plate and incubated for 2 h at 37 °C. To determine the phagocytic index (target cells ingested divided by the total number of macrophages multiplied by 100), flow cytometry was used.

**Stimulation and inhibition of the TNF pathway.** $1.5 \times 10^5$ MCF7 and MCF10 cells were seeded in a 24-well plate and stimulated with 100 or $1,000 \, ng \, ml^{-1}$ of recombinant human TNF-α (PeproTech). The following day 0.5 μM of the IKK2 (IκB kinase inhibitor), TPCA-1 (Tocris Bioscience), was added 4 h after TNF-α addition. CD47 expression was analysed by flow cytometry 48 and 72 h after TNF-α treatment. To block the binding of TNF-α to its receptor, $3 \times 10^5$ MCF7 and $3 \times 10^5$ MCF7-NFKB1shRNA cell lines were seeded in a six-well plate. The next day cells were stimulated with TNF-α ($100 \, ng \, ml^{-1}$) for up to 2 days. To block the TNF-α interaction with the TNF receptor, cells were treated with $100 \, \mu g \, ml^{-1}$ of a blocking monoclonal antibody (infliximab, Janssen), 1 h before TNF-α treatment. Effects on CD47 expression were analysed by flow cytometry 24 and 48 h after TNF-α treatment. Phagocytosis assays were performed as described above, 48 h after TNF-α treatment.

**ER stimulation.** β-Estradiol (Sigma Aldrich) was kindly provided by Maider Zabala Ugalde. MCF7 cells were grown in DMEM without phenol-red (Gibco), supplemented with 10% charcoal-filtered fetal bovine serum (FBS) (Gibco) and 1% pen/strep. $1.5 \times 10^5$ cells were seeded in a 24-well plate. The next day, half the cells were serum starved overnight and 2 days later all cells were stimulated with 100 nM β-Estradiol (E2). CD47 expression was analysed by flow cytometry and qPCR 24 and 48 h after E2 treatment. Presence or absence of serum had no effect on levels of CD47 expression on live cells; however, cells that were serum starved had a reduction on cell viability. Thus, in this study we only show the data obtained from cells that were not serum starved.

**Competition binding assay.** For the protein-DNA binding profiling assay, nuclear extract from either MCF7 or HePG2 cells was incubated with biotin-labelled oligos corresponding to an array of 48 ( − 1 used as a negative control) consensus sites for well-characterized transcription factors. Unbound oligo probes were washed and transcription factors-oligo probes complexes were hybridized to a plate (Signosis Inc). The captured probes were detected with streptavidin-HRP and a chemiluminescent substrate. To detect luminescence, a SpectraMax M3 Microplate Reader was used. Integration time was set to 1 s with no filter position. Binding of transcription factors to the corresponding consensus oligo probes was competed by incubating the nuclear extract with the CD47 E5 enhancer and the biotin-labelled consensus site oligos simultaneously. Thus, binding of each transcription factor to E5 results in reduced transcription factor-oligo probe complexes leading to lower luminescence detection. In this study, binding to E5 was calculated by taking the average of the relative luminescence units (RLU) produced by the binding of each transcription factor to the consensus probes when outcompeted with the E5 DNA fragment (binding competition) and subtracting this number from the average RLU produced by the binding of each transcription factor to the consensus sites probes only (control). Thus, binding of transcription factors to the E5 DNA fragment and not to the consensus sites probes is represented by an increase in RLU while not binding to E5 and binding to the consensus sites probes is represented by a no change or decrease in RLU. In some instances, we noticed for transcription factors that do not bind to E5 (for example, FAST-1, HIF, NF-E2, Stat1) a large decrease in RLU. It appears that addition of the CD47 E5 DNA fragment can promote more binding to the consensus oligo probes of some of E5 non-binding transcription factors (resulting in a substantial RLU decrease). Perhaps transcription factors-CD47 E5 bound complexes are not interfering with the 'non-binding transcription factors' allowing them to freely complex with their corresponding consensus probe.

**shRNA knockdown.** Candidate transcription factors were knockdown using TRIPZ inducible lentiviral shRNA constructs (ThermoFisher). Sense sequences within hairpins used to knockdown each of the candidate factors identified in the binding assay, are as follows:
STAT3: 1_ AAGTTCATGGCCTTAGGTA 2_GGCGTCCAGTTCACTACTA 3_TGACTTTGATTTCAACTAT
STAT5A: 1_GGCACATTCTGTACAATGA 2_CTGTGGAACCTGAAACCAT 3_TGGCTAAAGCTGTTGATGG
STAT5b: 1_CACCCTAATTTGACATCAA_
STAT6: 1_GCCTCTCTGACATATGCTA 2_GTTACTAGCCTTCTTCTCA 3_AGGAGACCACTGGAGAGCT
NFKB1: 1_ GCCCATACCTTCAAATATT 2_CAGGTATTTGACATATTAA 3_CCACAGATGTTCATAGACA
PPARα: 1_CGGACGATATCTTTCTCTT 2_TGAAGAGTTCCTGCAAGAA 3_CTTCTAAACGTAGGACACA. TurboRFP empty vector was used as an shRNA control.

MCF7 cells were seeded in six-well plates overnight and transduced the next day with each construct for 24 h. Transduced cells were selected with puromycin. To induce the shRNA and turboRFP, doxycycline was added to the cells media for 48 h. The efficiency of the knockdown was quantified by qPCR, and the effect on CD47 expression was analysed by qPCR and flow cytometry.

**BRD4 inhibition.** MCF7 cancer cells were seeded at $5 \times 10^5$ in a six-well plate overnight. The next day, cells were treated with 200 nM to1 µM JQ1 dissolved in DMSO, 500 µM PFI-1 (Selleckchem) dissolved in DMSO or 100 µM I-BET151 (Selleckchem) dissolved in EtOH. The following day, MCF7 cells were washed and the effect on *CD47* expression was analysed by qPCR over a period of 1 week for JQ1 and 2 days for PFI-1 and I-BET151. Jurkat cancer cells were treated with JQ1 following the same procedure performed for MCF7 cells. Jurkat cells were washed the next day after JQ1 treatment and effect on *CD47* expression analysed by qPCR over a period of 48 h. For this experiment, we used DMSO as a vehicle.

**Xenotransplants.** MCF7-GFP/Luc (GFP/Luciferase) cell line (kindly provided by Mingye Feng) was transduced with either turboRFP-NFKB1 shRNA or turboRFP empty vector (control). Then, cells were suspended in DMEM media containing 25% Matrix Matrigel (Becton Dickinson 354248). Cells were injected into the mammary fat pad of 4-8-week-old NOD.Cg-Prkdcscid Il2rgtm1Wjl/SzJ (NSG) female mice ($N = 5$). Tumours were allowed to grow, and a week later, doxycycline (100 µg) was injected into the mammary fat pad to induce the shRNAs and turboRFP. We followed tumour growth in individual mice for a period of six weeks, by measuring luciferase intensity.

**Patient-derived xenografts.** To grow patient-derived primary tumour samples, primary breast tumours obtained from informed and consented human subjects were dissociated as described below, lineage (CD3, CD16, CD31, CD45 and CD61)-depleted by flow cytometry and injected into cleared fat pads of 8-week-old female NSG mice ($N = 7$) as previously described[57].

**Animal models.** The NSG mice used for this study were maintained at the Stanford Animal Facility in accordance with the guidelines of the Administrative Panel on Laboratory Animal Care use committee (APLAC).

**Human samples.** Tissue specimens were obtained from consented patients as approved by Stanford University Institutional Review Board (IRB) protocol #350.

**Tumour dissociation.** Specimens were mechanically and enzymatically dissociated in DMEM/F12 media supplemented with 2% bovine serum albumin (BSA) containing collagenase hyaluronidase (Stem Cell Technologies) and DNAse (Worthington). Tissue was incubated for 16 h at 37 °C until single cell suspension was achieved. The single cell suspension was washed and treated with ACK (ammonium-chloride potassium) to lyse red blood cells, then washed and treated with Dispase and DNAse. Cells were filtered through a 40-micron filter, counted and stained with fluorescent antibodies for flow cytometry analyses.

**ChIP-Seq.** PDX breast tumours were dissociated as described above. Cells were resuspended in Hanks Balanced Salt Solution (Thermo Fisher) supplemented with heat-treated 2% FBS. Live cells were isolated using Ficoll-Paque Plus (GE Health Care Life Science) and counted. Due to a low number of cells recovered, bulk populations from samples PDX2 and PDX3 were submitted to ChIP-Seq. Samples PDX1 and PDX4 yielded enough number of cells allowing for the sorting of the tumorigenic population (that is, CD45 and H-2Kd negative, EpCAM^Hi and CD49f^Hi) prior ChIP-Seq. $1 \times 10^6$ cells were resuspended in 0.5 ml chromatin digestion buffer (33 mM Tris-acetate, pH 7.9, 66 mM potassium acetate, 10 mM magnesium acetate, 0.25% TX-100, 1 mM EGTA, 10 mM sodium butyrate) mixed with protease inhibitors (Roche Complete, EDTA free) and 15 µl of enzyme mix (equal volume each Fast Digest SaqAI, BfaI, Csp6I and NdeI from Thermo Fisher/Finnzymes). Chromatin digestions were performed at 37 °C for 30 min, followed by dilution with 0.5 ml 2X ChIP dilution buffer (220 mM KCl, 50 mM Tris-acetate, pH 7.9, 0.2% Sarkosyl, 0.2% Na-deoxycholate, 1.75% Tx-100, 40 mM EDTA, 1 mM EGTA). Debris were removed by centrifugation at 10,000 g for 10 min. The chromatin solution was transferred to a new tube, precleared with 40 µl Protein A-Dyna Beads for 2 h. chromatin solution (50 µl) was saved for Input. Anti-H3K27ac antibody (2 µg) (Active Motif #39133) and 40 µl Protein A-Dyna beads were added to remaining 0.95 ml of chromatin. Samples were then incubated overnight at 4 °C with rotation, in 1.5 ml tubes. Bead bound chromatin was washed twice with 1 × ChIP dilution buffer, once with 1 × ChIP dilution buffer (at 0.4 M NaCl), twice with Tris-EDTA (TE). DNA was eluted by resuspending beads in 100 µl TE containing 1% SDS and 10 mg Proteinase K, incubated at 65 °C for 1 h, with mixing every 15 min. Eluted DNA was column purified with Zymo DNA clean and concentrator kit in a final volume of 25 µl. ChIP-enriched DNA was quantified using a Qubit 3.0 and dsDNA HS assay. ChIP-enriched DNA (10–20 ng) was used to construct next generation sequencing libraries with NEXTERA. Indexed libraries were sequenced by NextSeq500 single end sequencing (75 bp).

**Flow cytometry analysis.** For staining, $1 \times 10^5$–$10^6$ cells were incubated with indicated antibodies (1:50–1:200) in cell sorting buffer (PBS with 2% FBS) on ice for 30 min. Cells were washed twice and analysed by fluorescence-activated cell sorting. Flow cytometry analyses were performed using a BD LSRFortessa. The antibodies used for this study were the following: PE anti-CD47 clone: B6H12 (BD 556046), APC anti-CD47 clone: B6H12 (eBioscience), PE anti-TNF RI/TNFRSF1A clone: 16803 (R&D Biosystem), APC anti-CRT (US Biological), PE-Cy7 anti-CD49f, clone: GoH3 (Biolegend), FITC anti-EpCAM, clone: 9C4c (Biolegend), Pac Blue anti-CD45, clone: HI30 (Biolegend), Pac Blue anti-H-2Kd, clone: SF1-1.1 (Biolegend). To quantify CD47 expression in different cancer cell lines, CD47-PE antibody bound per cell (ABC) was estimated by using phycoerythrin (PE) conjugated beads (QuantiBRITE, BD bioscience) and flow cytometry following the company's protocol.

**Protein extraction and western blotting.** Total protein was extracted from cultured MCF7 cells (TNF-α treated or untreated) using cell lysis buffer (Cell Signaling) supplemented with 1 × Halt Protease and Phosphatase Single-use Inhibitor Cocktail (Thermo Scientific) as recommended by the manufacturers. Western blots were performed as previously described[49]. IkBα (1:1,000) rabbit polyclonal, Phospho-IkBα (1:1,000) rabbit monoclonal and Tubulin (1:1,000) rabbit monoclonal (Cell Signaling) antibodies were used for detection of the proteins.

**Cell-death assay.** MCF7 cells ($5 \times 10^5$) were treated with 1 µM STS for 24 h. Cells were collected from the media and detached using TrypLe (Thermo Fisher). After detachment, cells were washed twice and resuspended in binding buffer (10 mM HEPES/NaOH, pH 7.4, 140 mM NaCl, 2.5 mM CaCl$_2$) with Annexin-V-AF488 antibody and incubated for 15 min on ice. Next, cells were washed twice and resuspended in binding buffer containing DAPI, and analysed by flow cytometry. Bivariant analysis of AF488-fluorescence (Annexin V) and DAPI-fluorescence marked the different cell populations.

**Quantitative real-time PCR (qPCR).** Total RNA was extracted using RNeasy Plus kits (Qiagen). cDNA was reversed transcribed using SuperScript III First-Strand Synthesis SuperMix (Invitrogen) and then amplified on the 7900HT Fast Real-Time PCR System (Applied Biosystems). Specific primers designed to amplify the desired gene were combined with cDNA and power SYBR Green (Thermo Fisher) following the manufacturer's instructions.
Primers used in this study are:
CD47
_F′:CATGGCCCTCTTCTGATTTC
_R′:GGAGGTTGTATAGTCTTCTGATTGG
NFKB1
_F′: CCCAGTGAAGACCACCTCTC
_R′: GGCACCACTGGTCAGAGACT
PPARα
_F′: CTGGAAGCTTTGGCTTTACG
_R′: CAATGCTCCACTGGGAGACT
STAT3
_F′: ACTAAGCCTCCAGGCACCTT
_R′ CGGACTGGATCTGGGTCTTA
STAT5a
_F′: GAGAAGTTGGCCGAGATCAT
_R′: GGTCACCAGGGCTGAGATAA
STAT5b
_F′: AAGATCAAGCTGGGGCACTA
_R′ CGGACCAACCTCTGTTCATT
STAT6
_F′: AGATGAGCCTGCCCTTTGA
_R′: GAGGAAACGCTGTCACTGG
TFF1
_F′: AATTCCTTCCAGCGCAAC
_R′: CACCATGGAGAACAAGGTGA
EGFP
_F′: ACTACCTGAGCACCCAGTCC
_R′: CTTGTACAGCTCGTCCATGC
β-actin
_F′: TCCCTGGAGAAGAGCTACG
_R′: GTAGTTTCGTGGATGCCACA

**Data availability.** The dbGAP accession number for the PDX breast tumour H3K27ac ChIP-Seq data sets generated in this study is phs001264. GEO database accession numbers for publicly available H3K27ac ChIP-Seq data sets analysed in this study are listed in Supplementary Table 1. All other relevant data are available from the corresponding authors on request.

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

## Acknowledgements

We want to thank J. Bradner for the inhibitor JQ1; the CD47 Disease Team for the CD47 antibody clone: Hu5F9-G4; M. Feng for the MCF7-GFP/Luc cell line and CRT antibody; S. Karten for critical review of the manuscript; P. Lovelace, J. Phuong-Nam Cheung, T. Storm, L. Jerabek, T. Naik and A. McCarty for technical assistance; A. Rada-Iglesias, E. Calo, S. Cai, M. Tal, A. Newman and D.B. Villar for scientific discussions. This project was supported by the Virginia and D.K. Ludwig Fund for Cancer Research; the National Cancer Institute of the National Institutes of Health under award number P01 CA139490 (to I.L.W.); the Immunology Training Grant Postdoctoral Fellowship 5 T32 AI07290-27 and the Cancer Research Institute Fellowship (to P.A.B.).

## Author contributions

P.A.B., B.J.A. and I.L.W. designed the research. P.A.B., B.J.A., T.S. and R.I.S. performed computational analyses. P.A.B., Y.Y.Y., S.B.W., F.K., M.Z., A.H.K., K.M., Y.K., P.H. and P.G. carried out the research. P.A.B., B.J.A., S.B.W., M.Z., A.K., Y.K., N.J.L., T.S., R.I.S., G.S. and M.F.C. contributed new reagents and tools. P.A.B., B.J.A., N.J.L., R.A.Y. and I.L.W. wrote the manuscript. All the authors reviewed, edited and approved the manuscript.

## Additional information

**Competing interests:** I.L.W. is co-founder, holds equity and is a Director and consultant of Forty Seven, Inc, a company developing CD47 antibody therapies. All other authors declare no competing financial interests.

