## [Peer Review File · Nature Communications]

Reviewers' comments:

Reviewer #1 (Remarks to the Author):

The manuscript describes the identification of super-enhancers for CD47 gene expression, which appear to be specific for certain cancer cell types, but not present in benign cells. ChIP-seq data and professional bioinformatics has been exploited to define genomic regions that can be scored as super-enhancers and these data were correlated with CD47 expression levels. Candidate enhancer elements were cloned into EGFP reporter constructs showing that certain enhancers acted in a cell-type specific manner. Specific transcription factors binding to one of these enhancers (E5) were identified in an elegant manner by competitive protein-DNA binding assays using a commercial plate-based TF-array and breast cancer cell extracts in comparison to a negative control cell line. However, statistics is missing for this assay - and it is not clear on which basis the TFs for further analysis have been selected. The enhancer region E5 has been studied in more detail revealing an NF- κ B binding site as potential effector site. This analysis also identified estrogen receptor binding sites, which have not been analyzed in more detail thereafter and which should be tested for potential functional roles. The effect of NF- κ B and inflammatory pathways on CD47 expression and phagocytosis was investigated by gene knockdown of NFKB1, experiments with TNF α , as well as anti-TNF antibodies in combination with anti-CD47 antibodies. Overall, the manuscript is well written and clear. It describes a very interesting study with a high level of originality and interest for the scientific community. However, several points should be addressed before publication:

Major points:

1. For the identification of transcription factors binding to E5 (Fig. 3): has any statistics been performed? Error bars are missing. If replicates are available, please show the statistics, if not: please repeat the experiment with triplicates.
Other TFs than the ones listed in the text showed a strong difference between MCF7 and HepG2, as well (e.g. TR, TCF...) - but have not been investigated in more detail. For some TFs (NF-E2, Stat1) a strong increase of the signal is observed specifically in MCF7 cells (instead of a decrease) upon addition of the CD47-E5: What is the interpretation for that? The authors should comment on that.
2. The identification of TF binding sites in E5 by PIQ is not explained in sufficient detail. Have DNase-seq experiments been carried out? The text implies that only computational methods of PIQ have been applied. What was then the DNase-seq-data that was used? Please clarify.
3. ESR1 and ESR2 sites identified by PIQ (Fig. 4a): have they been tested with the TF array (do they reflect ER in Fig. 3)? Please make that clear. A deletion construct eliminating these two sites should be tested (as in Fig. 4b), to elucidate whether estrogen receptor has a role in the E5 enhancer.
4. Flow cytometric analysis of macrophage/target cell coincidence (Fig. 5c) does not prove internalization (could also be binding). Authors might think of strengthening the phagocytosis assays by measuring luciferase activities at different time points after combining macrophages and MCF7-Luc cells infected with NFKB1-shRNA or control-shRNA (+/- CD47 antibody) - as the luciferase activity should decline over time when the phagocytosed MCF-7 cells are digested in phagolysosomes.
5. Fig. 5c: The pronounced synergistic effect of NFKB1-knock-down and CD47-antibody on the phagocytic index indicates that additional factors besides NF- κ B are responsible for CD47 expression. Given the ESR1 and ESR2 sites in the E5 of CD47 it would be interesting to knock down estrogen receptors similar to NFKB1 and measure the effects on phagocytosis of MCF-7 cells.
6. Fig. 5d: The postulated effect of BRD4 should also be tested with additional BET inhibitors such as I-BET-762 or PFI-1 - or with shRNA-mediated knockdown of BRD4
7. There are several NF- κ B binding sites in the promoter region of CD47 close to the transcription start site. Thus, the effect of TNF might be due to binding of NF- κ B to these sites instead of an effect on the more distal E5 enhancer. The E5-EGFP MCF7 reporter cells used in Fig. 2 should be used to determine an effect of TNF on the E5 enhancer. In the same experiment, a potential effect of estrogen should be tested.

Minor points:

- some typographic errors
- Fig. 5a: y-axis should be described in more detail: photon flux of luciferase activity? The data can be plotted as percentage of the initial Luc activity - which would allow to calculate mean values and error bars.
- E3 and E5 should be indicated in Fig. 1.
- The H3K27ac ChIP-seq data of HepG2 and MCF10 should be included in Fig. 1 if available.
- Fig. 6c: What are the histograms on the right side? (Left is labeled with CD47).
- The recent article of Sockolosky et al. should be discussed in the light of the present manuscript in the Discussion section (Sockolosky JT, Dougan M, Ingram JR, Ho CC, Kauke MJ, Almo SC, Ploegh HL, Garcia KC: Durable antitumor responses to CD47 blockade require adaptive immune stimulation. Proceedings of the National Academy of Sciences of the United States of America 2016, 113(19):E2646-2654.)

Johannes A. Schmid, Medical University Vienna, Austria

Reviewer #2 (Remarks to the Author):

Betancur et al. studied the regulatory mechanisms of CD47, a transmembrane protein that inhibits phagocytosis. Through analysis of the H3K27ac data, they identified two super-enhancers (SEs) that regulate the expression of CD47 in cancer cells. They confirmed that the SEs contain multiple active constituent enhancers. They then performed protein-DNA competitive binding assays and computational predictions to identify that NFKB1 binds to an constituent enhancer in MCF7 cells, which led to their investigation of the TNF α pathway's regulation on CD47 expression.

Inhibition of CD47 expressed in tumors is important to develop new therapeutic treatments. This is a very well designed study that thoroughly revealed the regulatory mechanisms of CD47 expression in breast cancer cell line with broad implication of CD47 regulation in other tumors and normal cells. The manuscript may be further enhanced by considering the following points.

1. In breast cancer cells, the downstream SEs associated with CD47 are only present in 2 out of 7 cell lines. The authors suggested that these SEs may be acquired in specific breast cancers. This speculation is certainly reasonable. Are these cell line specific SEs associated with any cell line specific mutations? Similarly, CD47 is regulated by different SEs in different cancer cells. Are these tissue-specific effect or the cancer-specific SEs are results of different mutations in different cancers?

2. A SE spans a large region by definition and contains multiple constituent enhancers. What is the relationship between these constituent enhancers? Are they collaborative? Mechanistic study may be out of the scope of this work but some discussion on this point would be helpful.

3. The active constituent enhancers were identified by checking the conservation across species and overlapping of H4K4me1/H3K27ac peaks. More details are needed to describe the parameters used to determine whether a region is conserved and how H3K4me1/H3K27ac signals are used to predict active enhancers and determine the enhancer regions.

3. Fig. 4, there are multiple predicted binding sites in E5. Are they all NFKB1 binding sites or they are predicted by PIQ as binding sites of other TFs? What are the parameters used in PIQ to predict these binding sites?

Minor:

1. In Fig. 1, label where E3.2, E5 and E7 are located in SEs (similar to Supplementary Fig. 2a). It would be much easier for readers to connect Fig. 1 and Fig. 2.

2. In Fig. 4, it would be useful to show the open chromatin signals and NFkB motifs if it does not make the figure too busy.

Reviewer #3 (Remarks to the Author):

To the authors:

The authors of the study "A super-enhancer associated with CD47 in breast cancer 2 link pro-inflammatory signaling to CD47 upregulation" show in their interesting manuscript that a super-enhancer regulates the "do not eat me" signal CD47 by NFkB. The authors show that these genetic switches turn on the transcription of this important gene. However there are several issues that should be addressed in a revision.

Novelty: The study is of particular interest.

Major comments:

- My major concern is the usage of cancer lines only, which is problematic. Is the observed effect specific for a limited number cancer cell lines? Is it possible for the authors to obtain a collection of patient derived tumor cells? Would it be possible to analyze these cells to substantiate the certainly interesting findings in the (artificial) cancer cell lines? E.g. correlate CD47 levels to NFkB activation; knock down of NFkB1 in these primary tumor cells and check for CD47 surface expression; etc...

- The in vivo experiments shown in Fig. 5 are interesting but incomplete. First it would be nice to show whether the shRNA has some influence (or not) on proliferation or cell survival/cell death in general in the MCF7 cells. Can the dramatic reduction in tumor mass be independent of CD47 down-regulation?

- TNF mediated upregulation of CD47 on MCF-7 cells is dependent on active signaling processes. Can the authors show that by increased activation of IKK or p65? Simultaneously Ikb phosphorylation and degradation should be tested in MCF7 and MCF10/HepG2.

Minor comments:

- The authors mention that specifically STATs , NFkB and PPAR are binding to E5. However the data also indicate that AR, COP, Oct4, PXR, TCF, TR, YY1 and especially NF1 are binding or at least competed by the E5 DNA (Fig.3). The authors should discuss that in detail.

- PPAR deletion with the shRNA was not very efficient (sFig.3). Can you repeat that experiment with a higher efficiency of PPAR knock down?

- In Fig. 5c the authors show a dramatic increase in phagocytosis. Is this due to the fact that the antibody is mediating facilitated phagocytosis ? But CD47 should be decreased on that cells. Do the authors have an explanation for that?

Reviewers' comments:

Reviewer #1 (Remarks to the Author):

The manuscript describes the identification of super-enhancers for CD47 gene expression, which appear to be specific for certain cancer cell types, but not present in benign cells. ChIP-seq data and professional bioinformatics has been exploited to define genomic regions that can be scored as super-enhancers and these data were correlated with CD47 expression levels. Candidate enhancer elements were cloned into EGFP reporter constructs showing that certain enhancers acted in a cell-type specific manner. Specific transcription factors binding to one of these enhancers (E5) were identified in an elegant manner by competitive protein-DNA binding assays using a commercial plate-based TF-array and breast cancer cell extracts in comparison to a negative control cell line. However, statistics is missing for this assay - and it is not clear on which basis the TFs for further analysis have been selected. The enhancer region E5 has been studied in more detail revealing an NF- κ B binding site as potential effector site. This analysis also identified estrogen receptor binding sites, which have not been analyzed in more detail thereafter and which should be tested for potential functional roles. The effect of NF- κ B and inflammatory pathways on CD47 expression and phagocytosis was investigated by gene knockdown of NFKB1, experiments with TNF α , as well as anti-TNF antibodies in combination with anti-CD47 antibodies. Overall, the manuscript is well written and clear. It describes a very interesting study with a high level of originality and interest for the scientific community. However, several points should be addressed before publication:

Major points:

1. For the identification of transcription factors binding to E5 (Fig. 3): has any statistics been performed? Error bars are missing. If replicates are available, please show the statistics, if not: please repeat the experiment with triplicates.

We want to thank the Reviewer for raising this important point. We repeated the experiment with a total of 4 replicates. We have now changed the way we present this data (Fig. 3a) with the aim of facilitating the interpretation. For the new figure, we averaged the relative luminescence units (RLU) produced by the binding of each transcription factor to the consensus probes when outcompeted with the E5 DNA fragment (binding competition). Next, we subtracted this number from the average RLU produced by the binding of each transcription factor to the consensus sites probes only (control). Thus, binding of transcription factors to the E5 DNA fragment and not to the consensus sites probes is represented by an increase in RLU while not binding to E5 and binding to the consensus sites probes is represented by a no change or decrease in RLU. The binding to E5 of each transcription factor obtained from the MCF7 nuclear extract

was compared to the binding to E5 of each transcription factor obtained from HepG2 nuclear extract, and these were significantly different by Student's unpaired t test for independent samples, $P < 0.01$ (**) and $P < 0.05$ (*). Error bars are now included.

Other TFs than the ones listed in the text showed a strong difference between MCF7 and HepG2, as well (e.g. TR, TCF...) - but have not been investigated in more detail.

We apologize to the Reviewer for the confusion surrounding this experiment. We hypothesized that in MCF7 cells CD47 is upregulated by the binding of transcription factors that work as activators to CD47 E5. Thus, we were searching for transcription factors that bound to CD47 E5 when MCF7 nuclear extract was used and did not bind when HepG2 nuclear extract was used. In the revised figure (Fig. 3a) we have attempted to make more clear that the factors TR, TCF and others do not fall into this category. For instance TR does not bind to CD47 E5 constituent when MCF7 nuclear extract is used and only slightly binds when HepG2 nuclear extract is used. On the other hand, TCF factors bind to CD47 E5 when either MCF7 or HepG2 nuclear extract is used. The methods and text have been revised to clarify how this experiment was designed.

For some TFs (NF-E2, Stat1) a strong increase of the signal is observed specifically in MCF7 cells (instead of a decrease) upon addition of the CD47-E5: What is the interpretation for that? The authors should comment on that.

In the original figure, an increase in signal when adding the CD47 E5 indicated more binding to the consensus probes (not binding to the CD47 E5). However, we agree this was confusing and we have now revised that figure to demonstrate increased binding to CD47 E5 with increased signal (with a decreased signal indicating binding to the consensus probes). However, we also noticed that in some instances addition of the CD47 E5 promoted more binding to the probes of transcription factors like HIF, NF-E2, Stat1, etc... that did not bind to CD47 E5. An explanation could be that transcription factors-CD47 E5 bound complexes are not interfering with the non-binding proteins allowing them to freely interact with the corresponding consensus probe. We now comment on this in the revised METHODS section.

2. The identification of TF binding sites in E5 by PIQ is not explained in sufficient detail. Have DNase-seq experiments been carried out? The text implies that only computational methods of PIQ have been applied. What was then the DNase-seq-data that was used? Please clarify.

We apologize for omitting a more detailed description of the computational approach we employed in this study, including our use of DNase-seq data publically available. We have now added the following specific details to the text

as well as the METHODS section of the paper: To locate the minimal region containing functional binding motif(s) within CD47E5_(1300bp), we first predicted regions within E5 that had motifs with binding capacity by using published Protein Interaction Quantification (PIQ) analyses. PIQ is an analytical method that combines existing ENCODE K562 DNase I hypersensitivity data ¹ with aggregate transcription factor specific scores obtained from JASPAR, UniPROBE and TRANSFAC to predict the probability of occupancy for a given candidate binding site in the genome by using standard parameters previously described ². The published resulting calls for putative binding motifs in the genome were visualized using the custom track feature from the UCSC genome browser (<https://genome.ucsc.edu>), with a score threshold of 800.

3. ESR1 and ESR2 sites identified by PIQ (Fig. 4a): have they been tested with the TF array (do they reflect ER in Fig. 3)? Please make that clear.

We apologize for not being clear. We clarify this with a new TF binding assay figure, which shows that ER does not bind to CD47 E5 when using nuclear extract from MCF7 but does bind at low levels to E5 when using nuclear extract from HepG2 (Fig. 3a). This result suggests that the ER does not directly upregulate CD47 expression, at least through the binding and activation of CD47 E5 constituent enhancer.

A deletion construct eliminating these two sites should be tested (as in Fig. 4b), to elucidate whether estrogen receptor has a role in the E5 enhancer.

We appreciate this suggestion made by the Reviewer, particularly given the extensive data that estrogen can stimulate tumor growth by activating genes involved in cancer progression. Although our competitive binding assays showed that in MCF7 estrogen did not bind to the predicted ER binding motifs within CD47E5 constituent, we have now specifically tested whether estrogen can regulate CD47 expression directly by interacting with the functional CD47 E5 constituent. To do so, we assayed the expression of endogenous CD47 and EGFP reporter signal of the E5-TK-EGFP construct in MCF7 after stimulating Estrogen Receptors (ER) by using β -Estradiol (E2). We did not observe a significant increase in CD47 expression or E5-TK-EGFP reporter signal by either QPCR or FACS after treatment. As a positive control we performed QPCR on the TFF1 gene, which has been shown previously to be upregulated upon ER stimulation ^{3,4}. These new data on ER stimulation are now included in the text, and provide important confirmatory data surrounding the specificity of the mechanism invoked herein (Supplementary Fig. 6d).

4. Flow cytometric analysis of macrophage/target cell coincidence (Fig. 5c) does not prove internalization (could also be binding). Authors might think of strengthening the phagocytosis assays by measuring luciferase activities at different time points after combining macrophages and MCF7-Luc cells infected

with NFKB1-shRNA or control-shRNA (+/- CD47 antibody) - as the luciferase activity should decline over time when the phagocytosed MCF-7 cells are digested in phagolysosomes.

We thank the Reviewer for this inquiry and suggestion. While the *in-vitro* phagocytosis assays used in this study are established and have been previously utilized in our lab and others⁵⁻⁸, we agree that macrophage/target cell coincidence by FACS does not necessarily prove internalization. Accordingly, we have now included images in the supplementary figures showing the overlap of target cells (TurboRFP+) with macrophages (GFP+) obtained using fluorescence microscopy. No overlap of GFP+ and TurboRFP+ cells is detected in the control shRNA panel when compared to the panel showing target cells treated with CD47 blocking antibody, NFKB1 shRNA or both. Moreover, in some of the cells the red fluorescence from the engulfed cells fades, presumably indicative of digestion by the phagocytic cell. We believe that these images would not be observed if the cells were simply doublets, and we have now added a comment to the revised figure legend to highlighting this point (Supplementary Fig. 4c).

5. Fig. 5c: The pronounced synergistic effect of NFKB1-knock-down and CD47-antibody on the phagocytic index indicates that additional factors besides NF-κB are responsible for CD47 expression. Given the ESR1 and ESR2 sites in the E5 of CD47 it would be interesting to knock down estrogen receptors similar to NFKB1 and measure the effects on phagocytosis of MCF-7 cells.

We agree with the Reviewer regarding the possibility that additional factors participate in the upregulation of CD47. Indeed, we have found that knocking down PPARα (a transcription that binds to CD47 E5 shown by our binding assay Fig. 3a) reduces CD47 expression (Fig.3b and Supplementary Fig 3c-d), confirming that cooperating factors may be involved at this locus. Regarding the specific question about the estrogen receptor, we have now found that stimulation of ER with E2 does not have an effect on CD47 expression in MCF7 (Please see inquiry 3.). These data suggest that knocking down ER will not have a CD47 mediated effect on phagocytosis of MCF7 cells. We agree that future studies will need to evaluate the combinatorial effect of other transcription factors at the CD47 SE, and have commented on this in the revised manuscript.

6. Fig. 5d: The postulated effect of BRD4 should also be tested with additional BET inhibitors such as I-BET-762 or PFI-1 - or with shRNA-mediated knockdown of BRD4.

As suggested by the Reviewer, we have tested additional BRD4 inhibitors I-BET151 and PFI-1 and confirmed that they too lead to a reduction in CD47 gene expression, as we had observed when using inhibitor JQ1 in MCF7 cells. We observed a significant decrease in CD47 transcript level in MCF7 cells beginning

at 24 hrs of treatment (Supplementary Fig. 5a,b). On the other hand, an increase in CD47 transcript levels was observed in the cell line HepG2 after 24 hrs of I-BET151 treatment (Supplementary Fig. 5c). These results confirm that the CD47 reduction we observe in MCF7 cells is achieved by directly affecting the binding of BRD4 to the breast cancer SE⁹. The upregulation of CD47 expression observed in the HepG2 cell line is probably achieved indirectly through the transcriptional down-regulation of CD47 inhibitors regulated by SEs.

7. There are several NF-κB binding sites in the promoter region of CD47 close to the transcription start site. Thus, the effect of TNF might be due to binding of NF-κB to these sites instead of an effect on the more distal E5 enhancer. The E5-EGFP MCF7 reporter cells used in Fig. 2 should be used to determine an effect of TNF on the E5 enhancer. In the same experiment, a potential effect of estrogen should be tested. Need to over express TNF on MCF7 CD47E5-EGFP cell line and measure GFP by flow.

We thank the Reviewer for raising these thoughtful points. To demonstrate that the expression of CD47 was upregulated directly by the CD47 E5 constituent enhancer responding to TNF stimulation and not to ER stimulation, we assayed the EGFP reporter signal of the E5-TK-EGFP construct in MCF7 cells after addition of TNF-α or E2. We observed an increase in EGFP transcript after 24 hours of TNF stimulation and an increase in protein levels at 24 hours, being more significant at 48 hrs after TNF stimulation (Supplementary Fig.6d upper panels). On the contrary, no effect or a slight reduction on CD47 expression was observed after stimulation of ER receptors at the transcript and protein levels (Supplementary Fig. 6d lower panels). These data extend recent work from our group showing that NFκB factors have a basal role in the activation of the CD47 promoter in atherosclerosis models¹⁰, but likely have another layer of specificity confirmed by the SE identified herein during the development of breast cancer.

Minor points:

- some typographic errors

- Fig. 5a: y-axis should be described in more detail: photon flux of luciferase activity? The data can be plotted as percentage of the initial Luc activity - which would allow to calculate mean values and error bars.

We have now made a new figure that includes calculated means, error bars and P values. Thank you for these important suggestions.

- E3 and E5 should be indicated in Fig. 1.

We included E5, E3.2 and E7 in Fig.1 as suggested by the Reviewer.

- The H3K27ac ChIP-seq data of HepG2 and MCF10 should be included in Fig. 1 if available.

H3K27ac data for HEPG2 is now included in Supplementary Fig. 1b, together with H3K27ac ChIP-seq data for other cancer cell lines lacking SEs. Unfortunately there is no publically available H3K27ac ChIP-seq data for MCF10.

- *Fig. 6c: What are the histograms on the right side? (Left is labeled with CD47). I don't understand.*

We apologize for not labeling this figure clearly. We corrected this and it should now be clear that the x axis is CD47 expression and the colored histograms are the different treatments. Gray histograms represent the Fluorescence Minus One (FMO) control.

- *The recent article of Sockolosky et al. should be discussed in the light of the present manuscript in the Discussion section (Sockolosky JT, Dougan M, Ingram JR, Ho CC, Kauke MJ, Almo SC, Ploegh HL, Garcia KC: Durable antitumor responses to CD47 blockade require adaptive immune stimulation. Proceedings of the National Academy of Sciences of the United States of America 2016, 113(19):E2646-2654.).*

We thank the Reviewer for this very interesting point and now discuss this paper in our revised DISCUSSION. We now discuss that although our main goal was to whether SEs regulate CD47 expression in cancer (and that their perturbation can affect at least one of the multiple functions this important molecule). It will be important to investigate the effect of reducing CD47 on other non-phagocytic functions of this gene. Clearly, understanding their role in immunocompetent models will be necessary to improve the CD47 targeting therapies that are currently being developed.

Reviewer #2 (Remarks to the Author):

Betancur et al. studied the regulatory mechanisms of CD47, a transmembrane protein that inhibits phagocytosis. Through analysis of the H3K27ac data, they identified two super-enhancers (SEs) that regulate the expression of CD47 in cancer cells. They confirmed that the SEs contain multiple active constituent enhancers. They then performed protein-DNA competitive binding assays and computational predictions to identify that NF κ B1 binds to an constituent enhancer in MCF7 cells, which led to their investigation of the TNF- α pathway's regulation on CD47 expression.

Inhibition of CD47 expressed in tumors is important to develop new therapeutic treatments. This is a very well designed study that thoroughly revealed the regulatory mechanisms of CD47 expression in breast cancer cell line with broad

implication of CD47 regulation in other tumors and normal cells. The manuscript may be further enhanced by considering the following points.

1. In breast cancer cells, the downstream SEs associated with CD47 are only present in 2 out of 7 cell lines. The authors suggested that these SEs may be acquired in specific breast cancers. This speculation is certainly reasonable. Are these cell line specific SEs associated with any cell line specific mutations? Similarly, CD47 is regulated by different SEs in different cancer cells. Are these tissue-specific effect or the cancer-specific SEs are results of different mutations in different cancers?

We thank the Reviewer for bringing up this important and very thoughtful question about the mechanism of enhancer formation near *CD47*. There are several models involving cancer mutations that could lead to an enhancer being formed at *CD47*, including 1) mutations within the enhancer itself, and 2) mutations in a protein-coding gene that is upstream of *CD47*. Either of these options could explain how SEs are acquired in a tissue-specific pattern during malignant transformation. A complete investigation of these hypotheses is difficult due to the limited availability of matched ChIP-seq and whole-genome sequence data from sets of healthy and malignant tissues. Alternatively, it is possible, if not likely, that epigenetic alterations of gene expression might result in development of super-enhancers. Here we investigated whether mutations could be involved in the higher expression of *CD47*, and if so, does a set of candidate mutations appear in the database?

1) Mutations in the enhancer itself

Enhancers can be nucleated by small mutations either in the germline or acquired somatically altering as a result transcription factor-binding sequences, so we investigated the DNA sequence at the *CD47* enhancer. We performed targeted sequencing of MCF7, four PDX breast tumor samples and HepG2 and found a variant in the E5 enhancer active in MCF7. This variant, a small insertion of "TTTGGGAC," is a sequence preferred by NFKB1 by motif analysis. This insertion is present in at least one allele in MCF7, and in three out of four PDX breast tumor samples and absent in HePG2 cells (which lack SEs and have low *CD47* levels). This, however (as shown below) is an ancient germline mutation, not a de novo somatic mutation in this cancer. Since the presence of the insertion is significantly correlated with enhancer activity, we cloned the insertion-containing and deletion-containing alleles into an EGFP-reporter construct and stimulated the TNF signaling pathway, which we show induces NFKB to bind to E5 to regulate *CD47* gene expression. When the insertion was present, the enhancer activity was significantly higher than when the insertion was absent [REDACTION]. This suggests that the insertion contributes to the regulatory activity of the enhancer. However, analyses of the 1000-Genome Project suggest this INDEL is a common variant: in ~2,500 individuals it was found that 37.1% had the insertion while 62.8% had the deletion. Thus, this is likely a germline

mutation, although we do not have normal cells from patient MCF7 to confirm that. It is possible that somatic mutations could promote SE formation, but are not the sole process which leads to their presence in cancers. Furthermore, since not all the PDX samples carrying the insertion are characterized by the presence of SEs, we investigated other possibilities by which SEs could be generated.

[REDACTION]

2) Mutations in a protein-coding gene that is upstream of CD47

Another way mutations in the genome could be responsible for the generation of the CD47 SEs could be by affecting the coding of a protein that is upstream of CD47, thus altering indirectly the CD47 regulatory landscape (e.g. by producing an imbalance of transcription factors and/or epigenetic markers that would be normally bound to CD47 enhancers for the activation of the gene). To check if we could find candidate mutated genes that could be responsible for the formation of SEs and if they were specific to a cancer type, we took advantage of publically information available at the IGGC (International Genome Cancer Consortium) database for tumors from donor patients and searched for genes mutated in breast and blood (cancer types where we have seen SEs, Figure 1a) and compared these mutated genes to mutated genes found in liver and colon cancer (cancer types where our two cell lines lack SEs, Supplementary Figure 1b). The graph below is a representation of the kind of information we can obtain by doing this comparison for the top 20 mutated genes found in breast (red arrows) and/or in blood (black arrows). We observed a pattern of mutated genes that are present specifically in either breast or blood cancers but not in both, but insofar as these data are from exome sequencing, they would not have shown the intergenic enhancer sequences that could have been informative. These data would require validation by testing each mutated gene and their ability to form SEs at the CD47 locus. Unfortunately, that is beyond the scope of this manuscript.

Top 20 Mutated Genes with High Functional Impact Simple Somatic Mutations (SSM)

2. A SE spans a large region by definition and contains multiple constituent enhancers. What is the relationship between these constituent enhancers? Are they collaborative? Mechanistic study may be out of the scope of this work but some discussion on this point would be helpful.

We thank the Reviewer for raising this question, and we now discuss this important point in the revised manuscript. As the Reviewer suggests, previous work has shown that in some cases most constituent enhancers within a SE have an additive effect and work in collaboration: deleting each of the constituents from the SE significantly decreases the expression of the target gene. However, in other cases, some constituent enhancers work best independently than in combination when tested in an enhancer-reporter assay¹¹. In our case, when we tested the TK-EGFP construct containing constituent enhancers E6 (which similar to E5 is also within the CD47 Super-enhancer) we did not notice a significant increase on EGFP reporter expression by FACS or fluorescence microscopy analyses, thus, implying that if this enhancer has regulatory activity, it might be cooperative. Further experiments are being performed to address this by using CRISPR knockdown systems to test whether deleting E5 and E6 in combination decreases CD47 expression to a greater

degree than when deleting the E5 alone. This work is outside the scope of the current manuscript, but we now comment in the discussion section the need to investigate a possible role for combinatorial effects of constituent enhancers at this locus.

3. The active constituent enhancers were identified by checking the conservation across species and overlapping of H4K4me1/H3K27ac peaks. More details are needed to describe the parameters used to determine whether a region is conserved and how H3K4me1/H3K27ac signals are used to predict active enhancers and determine the enhancer regions.

We apologize to the Reviewer for not giving sufficient details when describing the methods used to predict the active constituent enhancers. The following methods have now been added to the METHODS section: For many decades the study of enhancers has been based on the principle that non-coding genomic sequences that are conserved through evolution must be carrying important gene expression regulatory functions^{12,13}. To predict CD47 functional *cis*-regulatory regions (enhancers) within super-enhancers, we used The UCSC genome browser (<https://genome.ucsc.edu>) to identify highly conserved non-coding genomic regions across different species. To approach this, we aligned the human annotated CD47 genomic locus (~100Kb) to the CD47 annotated genomic locus available for a group of other vertebrate species including mouse, dog, armadillo, opossum, platypus, chicken, *Xenopus*, zebrafish and lamprey. In our analyses, the CD47 non-coding genomic sequence starts to diverge substantially in opossum and platypus, thus allowing for the visualization of isolated non-coding genomic patches of DNA that remained conserved across the different species compared (Supplementary Fig. 2a, black blocks). To identify highly evolutionary conserved genomic regions also marked by H3K27ac and H3K4me1 epigenetic modifiers (well characterized for marking open chromatin), we evaluated ENCODE H3K27ac and H3K4me1 ChIP-seq data sets available for a panel of 7 cell types and aligned them to the non-coding genomic CD47 sequences that were already aligned to find evolutionary conservation across different species (Supplementary Fig. 2a, colored peaks).

3. Fig. 4, there are multiple predicted binding sites in E5. Are they all NFKB1 binding sites or they are predicted by PIQ as binding sites of other TFs? What are the parameters used in PIQ to predict these binding sites?

We apologize to the Reviewer for omitting this information. By using PIQ, we predicted the exact location of binding sites for NFYA, ESRs, SPT23 and NFKB within E5. We have now added more details about PIQ in the METHODS section, including the following text: To locate the minimal region containing functional binding motif(s) within CD47E5_(1300bp), we first predicted regions within E5 that had motifs with binding capacity by using published Protein Interaction Quantification (PIQ) analyses. PIQ is an analytical method that combines existing

ENCODE K562 DNase I hypersensitivity data ¹ with aggregate transcription factor specific scores obtained from JASPAR, UniPROBE and TRANSFAC to predict the probability of occupancy for a given candidate binding site in the genome by using standard parameters previously described ². The published resulting calls for putative binding motifs in the genome were visualized using the custom track feature from the UCSC genome browser (<https://genome.ucsc.edu>), with a score threshold of 800.

Minor:

1. In Fig. 1, label where E3.2, E5 and E7 are located in SEs (similar to Supplementary Fig. 2a). It would be much easier for readers to connect Fig. 1 and Fig. 2.

Thank you for this suggestion. We included E5, E3.2 and E7 in Fig.1 and the associated figure legend as suggested by 2 of the Reviewers.

2. In Fig. 4, it would be useful to show the open chromatin signals and NFkB motifs if it does not make the figure too busy.

We have now added open chromatin signals to this figure as well as an explanation in the figure legend.

Reviewer #3 (Remarks to the Author):

To the authors:

The authors of the study "A super-enhancer associated with CD47 in breast cancer

2 link pro-inflammatory signaling to CD47 upregulation" show in their interesting manuscript that a super-enhancer regulates the "do not eat me" signal CD47 by NFkB. The authors show that these genetic switches turn on the transcription of this important gene. However there are several issues that should be addressed in a revision.

Novelty: The study is of particular interest.

Major comments:

- My major concern is the usage of cancer lines only, which is problematic. Is the observed effect specific for a limited number cancer cell lines? Is it possible for the authors to obtain a collection of patient derived tumor cells? Would it be possible to analyze these cells to substantiate the certainly interesting findings in the (artificial) cancer cell lines? E.g. correlate CD47 levels to NFkB activation; knock down of NFkB1 in these primary tumor cells and check for CD47 surface expression; etc...

We thank the Reviewer for this very valid and important point. To check if we could recapitulate what we have seen in the MCF7 cancer cell line, first, we tested for enhancers and potential SEs in patient derived-xenografted (PDX) breast tumor samples we could obtain. The tumors included (3) triple negative [negative for the expression of Her2, Progesterone (PR) and Estrogen (ER) markers] and (1) Progesterone Positive (PR+) primary breast cancer samples. Importantly, we found that in these primary tumors, the region downstream of CD47 (which includes E5 constituent) is enriched by Chip-seq with anti-H3K27Ac antibodies in a pattern similar to what we observed in the immortalized breast cancer cell lines. Moreover the H3K27ac enriched region for the PR+ sample (PDX1) qualified as a SE, resembling the MCF7 (PR+, ER+) SE originally identified in MCF7. These new data are shown in the revised Fig. 1b, and confirm that our findings are not confined to artificial cancer cell lines, but are also present in primary human tumor samples.

To further extend these studies, however, we also wished to test whether the primary human cancer cells could augment their CD47 expression levels in response to TNF- α stimulation, in a pattern similar to that observed in the immortalized cell lines. While these studies are difficult because these are not *in-vitro* adapted or selected cancer cells, and *in-vitro* incubation initiates the process of cell death, we did observe a 1.5 to 2.5 fold increase in the CD47 transcript and protein levels in 4 out of 5 samples after 24 hrs of TNF- α treatment (Fig. 6c and Supplementary Fig 6a). We next attempted to perform NFKB1sh, but due to a lack of long term cell viability in these culture, we could not complete these experiments. Ultimately, it will be important to analyze tens, or hundreds of breast cancer subtype samples for the presence of specific SEs, and in fact to analyze their cancer stem cells; such studies are beyond the scope of this current paper but have been highlighted as an area of future study in the discussion.

- The in vivo experiments shown in Fig. 5 are interesting but incomplete. First it would be nice to show whether the shRNA has some influence (or not) on proliferation or cell survival/cell death in general in the MCF7 cells. Can the dramatic reduction in tumor mass be independent of CD47 down-regulation?

We thank the Reviewer for this inquiry. We know that the shRNAs do not affect cell viability as no change in cell division rate was observed during 5 passages among the MCF7, MCF7 control shRNA and MCF7 NFKB1 shRNA conditions (Supplementary Fig. 4b). This was further confirmed by using an Annexin V-AF488 apoptosis detection assay with staurosporine as a control. There was no significant differences in the percentage of live, dead, necrotic or apoptotic cells between the three groups (Supplementary Fig. 4a). Thus, we conclude that the change in tumor growth observed herein appears to occur mainly by CD47 down-regulation.

- TNF mediated upregulation of CD47 on MCF-7 cells is dependent on active signaling processes. Can the authors show that by increased activation of IKK or p65? Simultaneously I κ B phosphorylation and degradation should be tested in MCF7 and MCF10/HepG2.

We thank the Reviewer for these thoughtful suggestions. To address these questions we blocked IKK2 by using its inhibitor TPCA-1 and assayed the effect on CD47 protein levels in MCF7, MCF10 and HepG2 cancer cells. We found that in cells incubated simultaneously with TPCA-1 and TNF- α , CD47 levels do not increase to the same levels observed when incubating the cells with the TNF- α ligand only (Supplementary Fig. 6b). By western blot analyses we also now show that in MCF7 cells phosphorylation and degradation of I κ B occurs after TNF- α stimulation. (Supplementary Fig.6c). Taken together, these new data help support the conclusion that NF κ B activation by stimulation of the TNF pathway likely specifically regulates CD47 expression.

Minor comments:

- The authors mention that specifically STATs , NF κ B and PPAR are binding to E5. However the data also indicate that AR, COP, Oct4, PXR, TCF, TR, YY1 and especially NF1 are binding or at least competed by the E5 DNA (Fig.3). The authors should discuss that in detail.

We thank the Reviewer for pointing this out. We realized that this graph was not very clear, thus, we repeated the experiments more times to be able to perform statistical analysis (as suggested by Reviewer 1) and also changed the presentation (Figure 3a). All the factors mentioned above by the Reviewer bind to E5 when using nuclear extract from MCF7. However, these factors also bind to E5 when nuclear extract from HepG2 is used. Since MCF7 1) expresses high levels of CD47 transcript and protein when compared to HepG2 and 2) MCF7 has SEs associated with CD47 while HepG2 does not, in this study we were interested principally in MCF7 factors binding to E5 (potential activators of CD47 expression) that did not bind to E5 in the HepG2 context (differential binding). These clarifications have been added to the revised manuscript.

- PPAR deletion with the shRNA was not very efficient (sFig.3). Can you repeat that experiment with a higher efficiency of PPAR knock down?

This point is very valid and we thank the Reviewer for the suggestion. We repeated the PPAR shRNA knockdown experiment by testing the 3 individual hairpins that were used in combination the first time we attempted this experiment. This time we increased the knockdown efficiency of PPAR α transcript from ~40% to ~75% when using PPAR shRNA hairpins 2 or 3. Both hairpins greatly reduced CD47 expression when compared to control shRNA or hairpin 1 (Supplementary Fig. 3c-e). This new result is very interesting and we

are continuing to investigate the role of the PPAR pathway in CD47 regulation. However, due to the time restriction and scope of this study, we did not discuss in detail the mechanism by which PPAR could be regulating CD47 in the revised manuscript.

- In Fig. 5c the authors show a dramatic increase in phagocytosis. Is this due to the fact that the antibody is mediating facilitated phagocytosis? But CD47 should be decreased on that cells. Do the authors have an explanation for that?

We thank the Reviewer for this question. First, it is important to mention that we also saw such synergistic effect on phagocytosis when treating the cells with Infliximab (the blocking antibody against TNF- α) and CD47 blocking antibody in this study. In another independent study we also saw the same synergistic effect after using blocking antibodies simultaneously against TNF- α and CD47 in a human smooth muscle and mouse atherosclerosis models¹⁰. Monocytes are known to increase the release of cytokines, including TNF- α as they differentiate into macrophages (Bener et al., 2016). The increase in phagocytosis effect we see when combining the NFKB1 shRNA and the CD47 blocking antibody treatment could be explained by a decrease of the CD47 transcriptional rate due to perturbation of the TNF pathway (either by NFKB shRNA or TNF- α blockage) which in turn slows the rate at which CD47 is localizing to the cell surface, thus permitting the blocking CD47 antibody to target more cells, rather than to target more CD47 molecules in a given cell.

Alternatively, it is possible that the observed increase in phagocytosis is a result of indirect regulation by NFKB1 or TNF- α on genes other than CD47, encoding antiphagocytic or prophagocytic signaling proteins. For instance, when we stimulated MCF7 cells with the TNF- α ligand, we also observed a modest increase in the expression of the prophagocytic signal, calreticulin, on the surface of MCF7 cells (Supplementary Fig. 7) We have now added these points to the discussion section of the manuscript.

References

- 1 Thurman, R. E. *et al.* The accessible chromatin landscape of the human genome. *Nature* **489**, 75-82, doi:10.1038/nature11232 (2012).
- 2 Sherwood, R. I. *et al.* Discovery of directional and nondirectional pioneer transcription factors by modeling DNase profile magnitude and shape. *Nat Biotechnol* **32**, 171-178, doi:10.1038/nbt.2798 (2014).
- 3 Laganier, J. *et al.* From the Cover: Location analysis of estrogen receptor alpha target promoters reveals that FOXA1 defines a domain of the estrogen response. *Proc Natl Acad Sci U S A* **102**, 11651-11656, doi:10.1073/pnas.0505575102 (2005).
- 4 Sun, J. M. *et al.* Estrogen regulation of trefoil factor 1 expression by estrogen receptor alpha and Sp proteins. *Experimental cell research* **302**, 96-107, doi:10.1016/j.yexcr.2004.08.015 (2005).
- 5 Jaiswal, S. *et al.* CD47 is upregulated on circulating hematopoietic stem cells and leukemia cells to avoid phagocytosis. *Cell* **138**, 271-285, doi:S0092-8674(09)00651-5 [pii] 10.1016/j.cell.2009.05.046 (2009).
- 6 Willingham, S. B. *et al.* The CD47-signal regulatory protein alpha (SIRPa) interaction is a therapeutic target for human solid tumors. *Proceedings of the National Academy of Sciences of the United States of America* **109**, 6662-6667, doi:10.1073/pnas.1121623109 (2012).
- 7 Zhang, H. *et al.* HIF-1 regulates CD47 expression in breast cancer cells to promote evasion of phagocytosis and maintenance of cancer stem cells. *Proc Natl Acad Sci U S A* **112**, E6215-6223, doi:10.1073/pnas.1520032112 (2015).
- 8 Zhang, M. *et al.* Anti-CD47 Treatment Stimulates Phagocytosis of Glioblastoma by M1 and M2 Polarized Macrophages and Promotes M1 Polarized Macrophages In Vivo. *PLoS One* **11**, e0153550, doi:10.1371/journal.pone.0153550 (2016).
- 9 Loven, J. *et al.* Selective inhibition of tumor oncogenes by disruption of super-enhancers. *Cell* **153**, 320-334, doi:10.1016/j.cell.2013.03.036 (2013).
- 10 Kojima, Y. *et al.* CD47-blocking antibodies restore phagocytosis and prevent atherosclerosis. *Nature* **536**, 86-90, doi:10.1038/nature18935 (2016).
- 11 Hnisz, D. *et al.* Convergence of developmental and oncogenic signaling pathways at transcriptional super-enhancers. *Mol Cell* **58**, 362-370, doi:10.1016/j.molcel.2015.02.014 (2015).
- 12 Cameron, R. A. *et al.* An evolutionary constraint: strongly disfavored class of change in DNA sequence during divergence of cis-regulatory modules. *Proc Natl Acad Sci U S A* **102**, 11769-11774, doi:10.1073/pnas.0505291102 (2005).
- 13 Cameron, R. A. & Davidson, E. H. Flexibility of transcription factor target site position in conserved cis-regulatory modules. *Dev Biol* **336**, 122-135, doi:10.1016/j.ydbio.2009.09.018 (2009).

REVIEWERS' COMMENTS:

Reviewer #1 (Remarks to the Author):

All my questions have been sufficiently addressed by the authors as specified in more detail in the attached document.

I recommend publication of the manuscript.

Johannes Schmid, Medical University of Vienna (Austria)

Note from the editor:

Reviewer 2 only commented for the editors. He/She found the revision adequate.

Reviewer #3 (Remarks to the Author):

dear authors

i do not have any further comments.

well done.